# Augmentation of myocardial $I_f$ dysregulates calcium homeostasis and causes adverse cardiac remodeling

Pessah Yampolsky [1,2,8], Michael Koenen[1,2,8], Matias Mosqueira [3,8], Pascal Geschwill[4,8], Sebastian Nauck [1,5], Monika Witzenberger [3], Claudia Seyler[1,5], Thomas Fink[1], Mathieu Kruska[1], Claus Bruehl[4], Alexander P. Schwoerer [6,7], Heimo Ehmke [6,7], Rainer H.A. Fink[3], Andreas Draguhn[4], Dierk Thomas[1,5], Hugo A. Katus[1,5] & Patrick A. Schweizer [1,5]

HCN channels underlie the depolarizing funny current ($I_f$) that contributes importantly to cardiac pacemaking. $I_f$ is upregulated in failing and infarcted hearts, but its implication in disease mechanisms remained unresolved. We generated transgenic *mice* ($HCN4^{tg/wt}$) to assess functional consequences of *HCN4* overexpression-mediated $I_f$ increase in cardiomyocytes to levels observed in *human* heart failure. $HCN4^{tg/wt}$ animals exhibit a dilated cardiomyopathy phenotype with increased cellular arrhythmogenicity but unchanged heart rate and conduction parameters. $I_f$ augmentation induces a diastolic $Na^+$ influx shifting the $Na^+/Ca^{2+}$ exchanger equilibrium towards 'reverse mode' leading to increased $[Ca^{2+}]_i$. Changed $Ca^{2+}$ homeostasis results in significantly higher systolic $[Ca^{2+}]_i$ transients and stimulates apoptosis. Pharmacological inhibition of $I_f$ prevents the rise of $[Ca^{2+}]_i$ and protects from ventricular remodeling. Here we report that augmented myocardial $I_f$ alters intracellular $Ca^{2+}$ homeostasis leading to structural cardiac changes and increased arrhythmogenicity. Inhibition of myocardial $I_f$ *per se* may constitute a therapeutic mechanism to prevent cardiomyopathy.

[1] Department of Cardiology, Medical University Hospital Heidelberg, Im Neuenheimer Feld 410, 69120 Heidelberg, Germany. [2] Department of Molecular Neurology, Max-Planck-Institute for Medical Research, Jahnstrasse 29, 69120 Heidelberg, Germany. [3] Division of Medical Biophysics, Institute of Physiology and Pathophysiology, Heidelberg University, Im Neuenheimer Feld 326, 69120 Heidelberg, Germany. [4] Division of Neuro- and Sensory Physiology, Institute of Physiology and Pathophysiology, Heidelberg University, Im Neuenheimer Feld 326, 69120 Heidelberg, Germany. [5] DZHK (German Centre for Cardiovascular Research) partner site Heidelberg/Mannheim, Im Neuenheimer Feld 410, 69120 Heidelberg, Germany. [6] Department of Cellular and Integrative Physiology, University Medical Centre Hamburg-Eppendorf, Martinistrasse 52, 20246 Hamburg, Germany. [7] DZHK (German Centre for Cardiovascular Research) partner site Hamburg/Kiel/Lübeck, Martinistrasse 52, 20246 Hamburg, Germany. [8] These authors contributed equally: Pessah Yampolsky, Michael Koenen, Matias Mosqueira, Pascal Geschwill. Correspondence and requests for materials should be addressed to P.A.S. (email: patrick.schweizer@med.uni-heidelberg.de)

The hyperpolarization-activated cyclic nucleotide-gated channel 4 (HCN4) is the dominant HCN-isoform in the sinoatrial node and is significantly involved in generation and regulation of heart rhythm[1–4]. Apart from its abundant expression in the pacemaker and conduction system, the adult working myocardium is characterized by low HCN levels[5,6]. At early embryonic stages, however, HCN4 is abundantly transcribed in the whole heart[6] and contributes importantly to $I_f$ triggered automaticity of ventricular myocytes[7]. Moreover, HCN4 was identified as a cell marker for the cardiomyogenic progenitor pool of the first heart field, implicated in the earliest stage of cardiac mesoderm formation and morphogenesis[8–10]. Toward birth HCN4 transcription is downregulated in working-type cardiomyocytes[6] and remains at low levels during adult stages, suggested to prevent pathological remodeling[6]. In this context, increased $I_f$ and HCN expression was reported in failing hearts and after myocardial infarction[5,11,12], but its pathophysiological role remains to be established. Notably, pharmacological blockade of $I_f$ improves cardiovascular outcome in patients with chronic heart failure—results that were considered to be driven mainly by improved myocardial energy supply at lower heart rates[13,14]. However, recent data demonstrated that $I_f$ blockade in patients with coronary artery disease with preserved myocardial function did not improve outcome despite marked reduction of heart rate[15], pointing towards mechanisms of $I_f$ blockade that may be particularly beneficial in failing hearts. In this regard, pleiotropic effects of $I_f$ blockade beyond heart rate reduction have been suggested[16,17], but these mechanisms have not been specified yet.

To study the significance of increased $I_f$ in the working myocardium, we generated transgenic mice that express human HCN4 (hHCN4) under control of the murine cardiac troponin I (cTNI) gene promoter. We here show that a moderate increase of $I_f$ in cardiac myocytes to levels observed in heart failure leads to cardiac dilation and impaired cardiac function. HCN4 activity essentially is linked to calcium regulation of cardiomyocytes and augmented $I_f$ results in dysregulated $Ca^{2+}$ homeostasis driving cell death and cardiac remodeling.

## Results

### Generation of transgenic mice overexpressing hHCN4.
To study the significance of HCN4 channel overexpression-mediated $I_f$ augmentation in the working myocardium, transgenic mice were generated that express hHCN4 under control of the murine cTNI gene promoter[18,19] (Fig. 1a). HCN4$^{tg/wt}$ offspring are born at a mendelian frequency and exhibit no overt cardiac or general abnormalities. HCN4$^{tg/wt}$ mice produce abundant hHCN4 transcripts (Fig. 1b) leading to increased HCN4 protein levels (Fig. 1c). Intense anti-HCN4 staining signals at the plasma membrane of HCN4$^{tg/wt}$ cardiomyocytes (Fig. 1d) indicate efficient trafficking of hHCN4 channels to the cell surface, while wild type tissue yields only little anti-HCN4 signals. Accordingly, patch-clamp recordings showed significantly increased $I_f$ densities in HCN4$^{tg/wt}$ cardiomyocytes. At physiological resting membrane potentials (RMP) of ventricular cardiomyocytes (~−90 mV) we observed a two–three3-fold higher $I_f$ density in HCN4$^{tg/wt}$ cardiomyocytes compared to wild type (Fig. 1e, f), recapitulating the magnitude of current increase that was observed in cardiomyocytes of heart failure patients[5].

### HCN4$^{tg/wt}$ mice develop structural heart disease.
While cardiac morphology was similar at birth, we uncovered cardiac dilation with significantly increased heart-to-body weight ratios in HCN4$^{tg/wt}$ mice 2 months postpartum (Fig. 2a–g). Right ventricular diameters, in particular, were found markedly enlarged (Fig. 2d) and dilation was accompanied with reduced wall thickness (Fig. 2e). To assess cardiac in vivo phenotypes, echocardiographic evaluations were performed at 3 and 6 months of age. At 3-month of age left ventricular wall thickness of HCN4$^{tg/wt}$ mice was lower compared to wild type (Fig. 2l). Similar to the histological findings at 2 months of age there was a trend toward higher left ventricular internal diameters in HCN4$^{tg/wt}$ animals (Fig. 2j, k), although these changes did not reach statistical significance. Fractional area shortening did not differ between groups at this age (Fig. 2i). At 6 month of age, however, HCN4$^{tg/w}$ mice displayed a dilated cardiomyopathy phenotype (Fig. 2h, i) with lower left ventricular wall thickness, dilation of left ventricular internal diameters (Fig. 2n–p) and systolic dysfunction with reduced fractional shortening of 22.9 ± 1.4% compared to 27.5 ± 1.4% in controls (means ± s.e.m., P = 0.031, unpaired t-test) (Fig. 2i). Furthermore mitral valve Doppler imaging displayed increased E/A ratios in transgenic mice indicative for impaired left ventricular relaxation and diastolic dysfunction (2.16 ± 0.22 in HCN4$^{tg/w}$ mice compared to 1.62 ± 0.07 in controls, P = 0.021, unpaired t-test) (Fig. 2m, q).

In line with macroscopic alterations in HCN4$^{tg/wt}$ hearts, we observed significantly increased diameters of cardiomyocytes (Fig. 2c) and elongated sarcomeres (Fig. 3a, b). However, HCN4 overexpression did not induce cardiac fibrosis (Fig. 3c, d), and we observed no myocyte disarray or cellular infiltrations (Fig. 3a, e). Early treatment of HCN4$^{tg/wt}$ mice with the $I_f$ blocker ivabradine significantly reduced chamber dilation and wall rarefication (Fig. 2b–g), and normalized cardiomyocyte diameters (Fig. 2c), indicating that increased $I_f$ causes the altered structural phenotype.

### Transcriptional alterations in HCN4$^{tg/wt}$ hearts.
We next asked whether structural and functional remodeling in hearts of HCN4$^{tg/wt}$ mice is associated with activation of the fetal gene program. Evaluation of mRNA, isolated from 6-months-old mice, revealed markedly increased levels of the myocardial cell growth markers GSK-3B and mTOR (Fig. 4a) and slightly upregulated levels of the prototypical hypertrophic marker genes MYH7 (encoding ßMHC), CnA (encoding calcineurin A) and NPPA (encoding ANP) (Fig. 4b). We further explored transcriptional profiles of ion channels and transporters, implicated in action potential generation, excitation-contraction coupling and intracellular calcium homeostasis (Fig. 4c, d). Interestingly, HCN4$^{tg/wt}$ hearts displayed significant downregulation of the sodium–calcium exchanger gene NCX1 (Fig. 4c) and intrinsic murine HCN genes were slightly diminished (Fig. 4d). Furthermore, levels of transcripts of the repolarizing channel genes KCNQ1 (encoding Kv7.1) and KCND2 (encoding Kv4.2) were decreased, while the repolarizing channel gene KCNH2 (encoding Kv11.1) showed a trend toward higher levels (Fig. 4e). Other genes regulating calcium handling and cellular electrophysiology, however, showed transcript levels that were not significantly changed.

### HCN4 overexpression impacts calcium handling proteins.
Based on NCX1 downregulation in HCN4$^{tg/wt}$ hearts, we hypothesized that dysregulated intracellular sodium–calcium homeostasis might be a pathomechanistic component in $I_f$ overexpression-mediated cardiomyopathy. To evaluate a contribution of calcium handling proteins we performed immunoblotting of various key players. In line with the transcriptional analysis we observed significantly reduced levels of NCX1 protein in HCN4$^{tg/wt}$ hearts, a finding that was paralleled by lower levels of SERCA2 and phospholamban (PLN) whole protein content (Fig. 4e, f). Of note, phosphorylation of PLN at its CaMKII-specific Thr-17 site was significantly increased, whereas the rise at

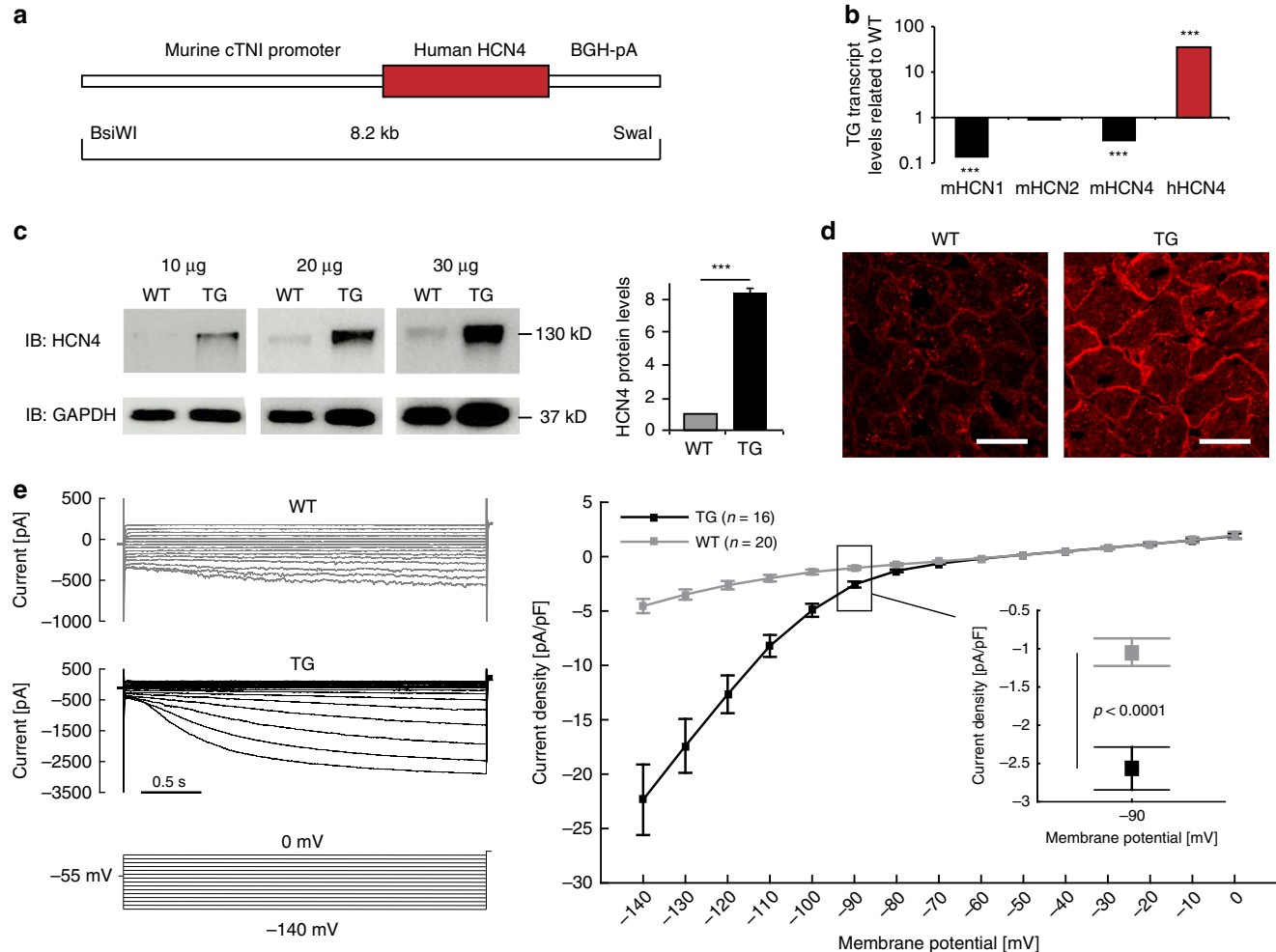

**Fig. 1** Generation and characterization of transgenic $HCN4^{tg/wt}$ mice. **a** The transgene carries a 4.3 kb promoter fragment of the murine cTnI gene (*cTnI*) fused to the *human HCN4* cDNA and the bovine growth hormone gene poly A signal (BGH-pA). **b** Profiling of the endogenous murine HCN1 (mHCN1), HCN2 (mHCN2), HCN4 (mHCN4), and transgenic *human HCN4* (*hHCN4*) transcripts in the ventricular myocardium of 2-month-old $HCN4^{tg/wt}$ mice measured by quantitative real-time PCR, (qRT-PCR). Wild type = 1.0 ($n = 6$ animals; ***$P < 0.001$; unpaired *t*-test). **c** Western blot and quantitative analysis of HCN4 protein in the ventricular myocardium of $HCN4^{tg/wt}$ transgenic and wild type *mice*, normalized to glyceraldehyde 3-phosphate dehydrogenase (GAPDH) for control of protein loading. For comparison samples with different protein load (10, 20, 30 μg) are shown ($n = 6$ animals/ groups; *$P < 0.05$, **$P < 0.01$, ***$P < 0.001$; unpaired *t*-test). **d** Immunohistochemistry for HCN4 (red) in ventricular myocardium of wild type (left) and transgene (right) *mice*. Scale bars 20 μm. Representative $I_f$ currents (**e**) and current-voltage relationship (**f**) recorded from ventricular cardiomyocytes isolated from $HCN4^{tg/wt}$ and wild type *mice* ($n = 20$ cells from six wild type *mice* and $n = 16$ cells from five $HCN4^{tg/wt}$ *mice*; *$P < 0.05$, **$P < 0.01$, ***$P < 0.001$; unpaired *t*-test). Data are expressed as mean ± s.e.m. Source data are provided as a Source data file

its protein kinase A (PKA)-site Ser-16 did not reach statistical significance. Moreover, the $Ca^{2+}$-sensor protein μ-calpain showed higher levels in $HCN4^{tg/wt}$ hearts, while the L-type calcium channel protein Cav1.2 remained unchanged (Fig. 4e, f). These data document profound alterations in the regulation of calcium handling proteins under the influence of augmented myocardial $I_f$.

**$HCN4^{tg/wt}$ cardiomyocytes show altered $Ca^{2+}$ homeostasis.** To determine the influence of increased $I_f$ on intracellular $Ca^{2+}$ handling, we measured $[Ca^{2+}]_i$ transients in isolated, electrically driven cardiomyocytes of wild type and $HCN4^{tg/wt}$ mice at baseline and after incubation with 3 μM ivabradine (Fig. 5a). Remarkably, we observed a distinctive rise in $[Ca^{2+}]_i$ baseline levels and significantly increased systolic $[Ca^{2+}]_i$ transients under steady state field stimulation of cardiomyocytes isolated from $HCN4^{tg/wt}$ hearts (Fig. 5a–d). The efficiency of $Ca^{2+}$ release, however, reflected by

the time to peak, was almost unchanged (Fig. 5e). Of note, $[Ca^{2+}]_i$ levels, high in $HCN4^{tg/wt}$ cardiomyocytes, dropped to significantly lower levels in ivabradine-treated $HCN4^{tg/wt}$ cardiomyocytes (Fig. 5b), suggesting that abundant HCN4 current mediates diastolic $Ca^{2+}$ overload and altered $Ca^{2+}$ homeostasis. Furthermore, the markedly increased slope of $Ca^{2+}$ uptake (Fig. 5f) and shortened time to half decay ($D_{50}$) of calcium transients (Fig. 5d) indicate that $HCN4$ overexpression influences uptake of increased $[Ca^{2+}]_i$ to the SR mediated by SERCA. Consistently, $I_f$ inhibition normalized the slope $Ca^{2+}$ uptake and $[Ca^{2+}]_i$ parameters in cardiomyocytes of transgenic animals (Fig. 5a–f). In addition, $Ca^{2+}$ load experiments using caffeine 10 mM (Supplementary Fig. 1) showed that the caffeine-induced $Ca^{2+}$ release peak was significantly higher in $HCN4^{tg/wt}$ cardiomyocytes, indicating that the SR $Ca^{2+}$ content is increased, which is consistent with the experimental data of higher $Ca^{2+}$ transients upon electrical stimulation (Fig. 5a–c). Of note, recordings of $Ca^{2+}$ transients

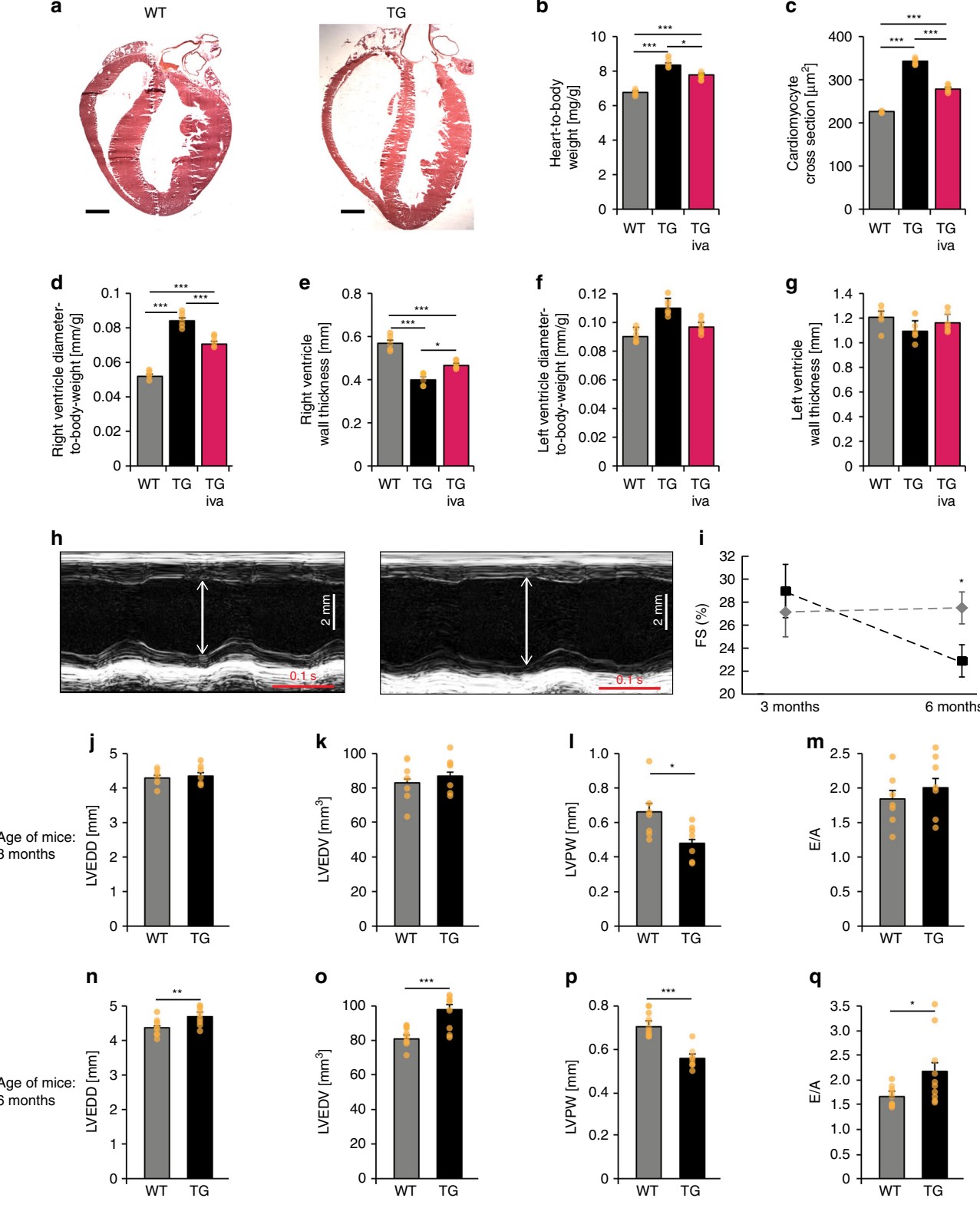

showed abnormal prolongation of diastolic $Ca^{2+}$ decay ($D_{90} >$ 2000ms) in 39% of cardiomyocytes isolated from HCN4$^{tg/wt}$ hearts (evaluated separately—please refer to Supplementary Fig. 2), changes that were reduced to wild type level after treatment of HCN4$^{tg/wt}$ cardiomyocytes with ivabradine, underlining that increased $I_f$ profoundly disorganizes cellular $Ca^{2+}$ homeostasis.

**Reverse mode NCX blockade ameliorates altered $Ca^{2+}$ cycling.** To elucidate the role of NCX in HCN4-mediated perturbation of $Ca^{2+}$ cycling we recorded $[Ca^{2+}]_i$ transients from wild type and HCN4$^{tg/wt}$ cardiomyocytes at baseline and in the presence of the selective NCX inhibitor ORM-10103[20].

Interestingly, NCX inhibition by ORM-10103 (10 µM) reversed the changes in $Ca^{2+}$ metabolism nearly to wild type

**Fig. 2** Morphological and functional phenotype of transgenic $HCN4^{tg/wt}$ mice. **a** Representative HE-stained cryosections display morphology of 2 months-old wild type (left) and $HCN4^{tg/wt}$ (right) hearts showing cardiac dilation and significantly reduced wall thickness in particular of the right ventricle. Scale bar 1 mm. **b–g** HE-stained cryosections illustrating quantitative effects of hHCN4 transgene overexpression on the structural phenotype of hearts in the presence and absence of ivabradine at two month postpartum. Heart-to-body weight ratio (**b**), cross section of left ventricular cardiomyocytes (**c**), right ventricle diameter-to-body weight ratio (**d**), and wall thickness (**e**), left ventricle diameter-to-body weight ratio (**f**) and wall thickness (**g**) of wild type, $HCN4^{tg/wt}$, and ivabradine-treated $HCN4^{tg/wt}$ mice at 2 months postpartum ($n = 6$ animals/groups; $*P < 0.05$, $**P < 0.01$, $***P < 0.001$; ANOVA). **h–q** Transthoracic echocardiography analyses of wild type (left) and $HCN4^{tg/wt}$ mice. The analyses were performed in 3 months- (**j–m**) and 6 months- (**n–q**) old mice. Representative echocardiographic M-mode views in 6 months-old mice (**h**), and quantification of fractional shortening (FS) (**i**), left ventricular end-diastolic diameters (LVEDD) (**j**, **n**), and volume (LVEDV) (**k**, **o**), posterior wall thickness in diastole (LVPW) (**l**, **p**) and E/A ratio (**m**, **q**) are depicted (3 months-old mice $n = 7$ wild type and $n = 7$ transgenic animals/group; 6 months-old mice $n = 12$ wild type and $n = 10$ transgenic animals/group; $*P < 0.05$, $**P < 0.01$, $***P < 0.001$; unpaired t-test). Data are expressed as mean ± s.e.m. Source data are provided as a Source data file

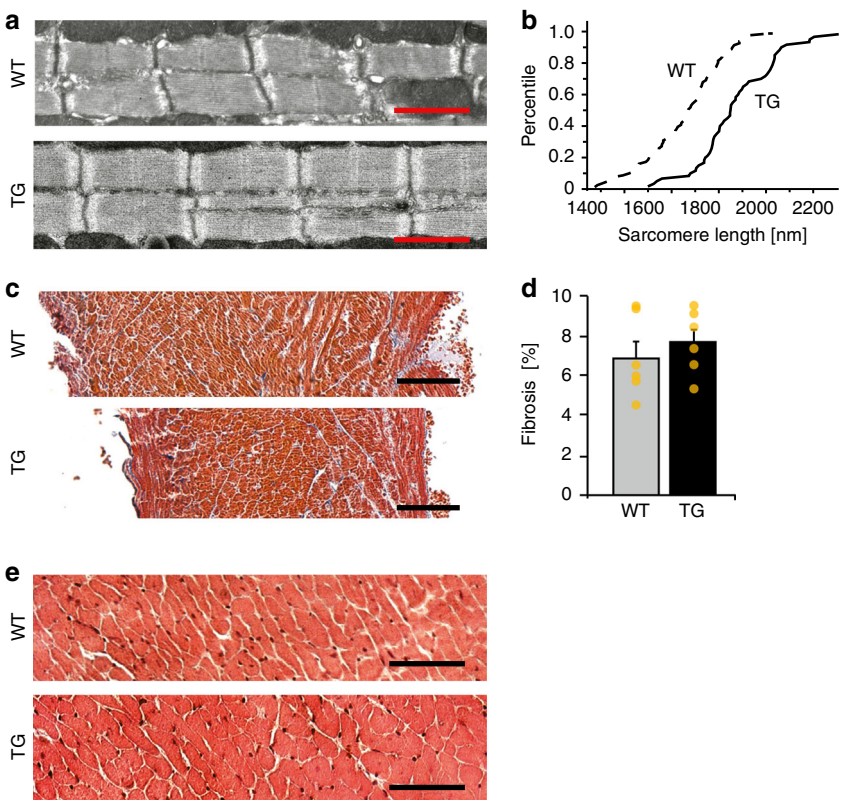

**Fig. 3** Histological analysis of transgenic $HCN4^{tg/wt}$ hearts. **a** Electron microscopy analysis of left ventricular sarcomere structure of 2-months-old wild type and $HCN4^{tg/wt}$ hearts. Scale bars: 1250 nm. **b** Comparative percentile plot of sarcomere lengths' distribution in cardiomyocytes of wild type (dashed line, $n = 480$ sarcomeres from six mice) and $HCN4^{tg/wt}$ (solid line, $n = 480$ sarcomeres from six mice) animals. **c** Masson's trichrome staining of right ventricular paraffin sections from wild type and $HCN4^{tg/wt}$ hearts, 2 months postpartum. Scale bars: 100 μm. **d** Comparative analysis of fibrosis by semi-automatic quantification using ImageJ-based plug-in showed similar levels of fibrosis between wild type (gray column) and $HCN4^{tg/wt}$ (black column) tissue ($n = 6$ animals/group; $*P < 0.05$, $**P < 0.01$, $***P < 0.001$; unpaired t-test). **e** HE staining of right ventricular cryosections from wild type and $HCN4^{tg/wt}$ hearts, 2 months postpartum. Scale bars: 50 μm. Data are expressed as mean ± s.e.m. Source data are provided as a Source data file

levels (Fig. 6a–f). Increased diastolic $[Ca^{2+}]_i$ declined remarkably (Fig. 6b) and the amounts of total cellular $Ca^{2+}$ movement, reflected by the peak area, returned to normal levels (Fig. 6c). Moreover, the increased slope $Ca^{2+}$ uptake (Fig. 6f) and shortened time to half decay ($D_{50}$) of calcium transients (Fig. 6d) were restored to levels not significantly different from wild type. Thus, our data demonstrate an important pathophysiological implication of the NCX current in the changed cardiac $Ca^{2+}$ homeostasis of $HCN4^{tg/wt}$ animals.

**$HCN4^{tg/wt}$ hearts show increased apoptosis.** We then asked whether increased $[Ca^{2+}]_i$ would provoke apoptosis in $HCN4^{tg/wt}$ hearts. Using TUNEL assay, we detected significantly higher proportions of nuclei with DNA fragmentation in ventricles of $HCN4^{tg/wt}$ hearts compared to wild type (Fig. 7a, b). In addition,

screening of genes associated with apoptosis by qRT-PCR demonstrated a significant rise of the apoptosis effector caspase-3 in transgenic myocardium (Fig. 7c, d; Supplementary Fig. 3). Moreover, we observed higher transcript levels of the $Ca^{2+}$-sensor protease μ-calpain and the tissue transglutaminase (Fig. 7d), both implicated in myocardial cell death and structural remodeling in association with $[Ca^{2+}]_i$ overload[21]. Further analysis of caspase-3 tissue activity revealed a marked increase of caspase-3-positive cells in $HCN4^{tg/wt}$ ventricles (Fig. 7e, f), pointing to increased apoptotic cell death as an explanation for wall thinning and dilated cardiomyopathy.

**Electrophysiological changes in $HCN4^{tg/wt}$ cardiomyocytes.** Action potential (AP) recordings of $HCN4^{tg/wt}$ and wild type cardiomyocytes were performed using ruptured whole-cell patch

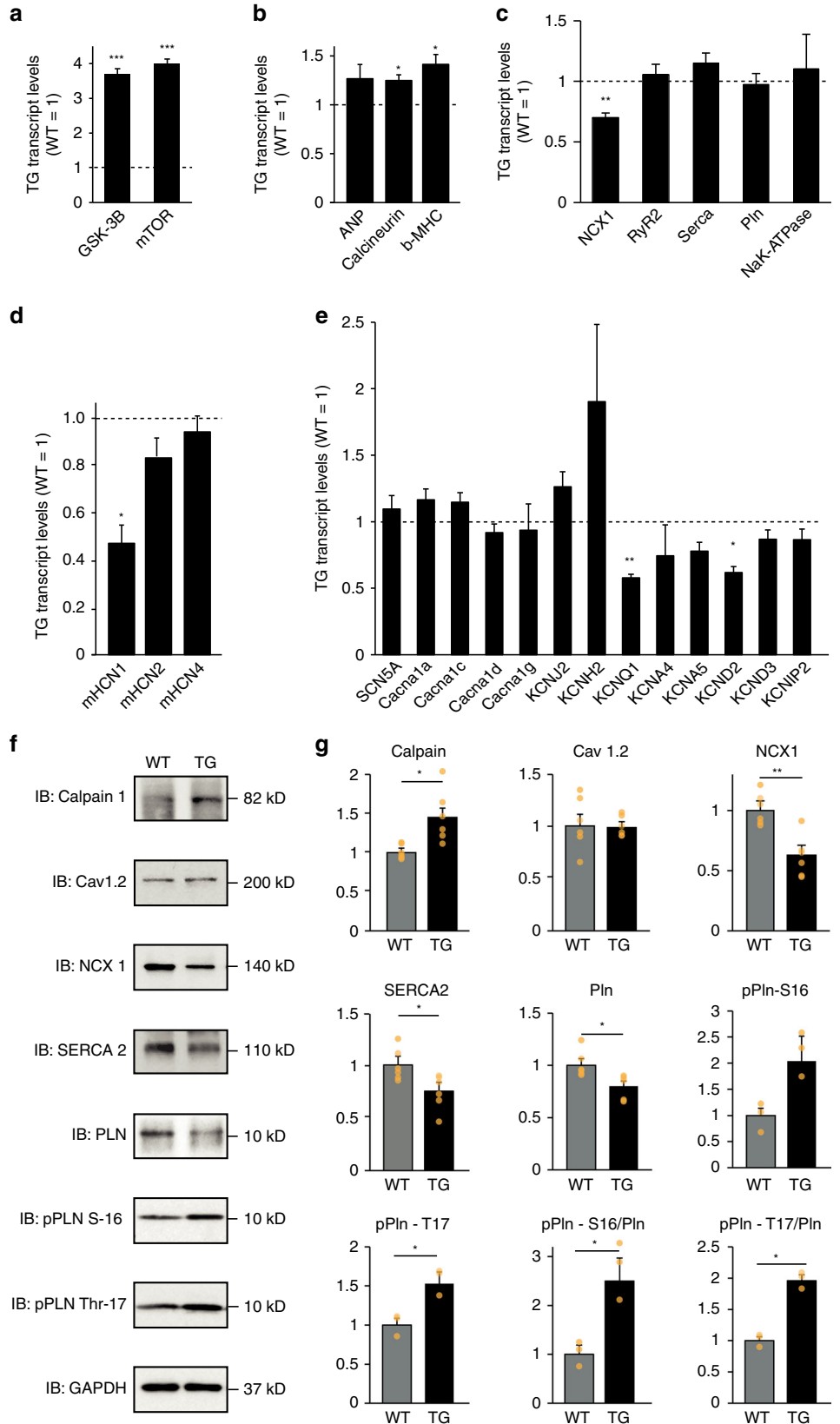

clamp technique (Fig. 8a). In HCN4$^{tg/wt}$ cardiomyocytes RMP was more positive than in wild type cardiomyocytes (wild type: $P_{50} = -61.25$ mV, $n = 14$ vs. HCN4$^{tg/wt}$: $P_{50} = -54.1$ mV, $n = 12$; $p = 0.0013$; unpaired $t$-test) (Fig. 8c), consistent with diastolic depolarisation caused by HCN4-mediated sodium influx.

Likewise, the amplitude of overshoot was diminished in transgenic cardiomyocytes (Fig. 8b), pointing to an inactivation of voltage-gated Na$^+$-current at more depolarized RMP. AP properties were evaluated at three different pacing frequencies (0.5, 1, 2 Hz) (Fig. 8d–f). AP duration at 20 (APD$_{20}$) and 50% (APD$_{50}$)

**Fig. 4** Transcriptional changes and expression of $Ca^{2+}$ handling proteins in $HCN4^{tg/wt}$ hearts. qPCR and Immunoblots from heart samples of 6 months-old *mice*. qPCR measurements of gene transcription related to **a** cell growth, **b** the fetal gene program and hypertrophy, **c** $Na^+/Ca^{2+}$-handling, **d** murine HCN channels, and **e** cardiac ion channels (6 months-old *mice* $n = 4$–8 animals/group). The bars show the transgenic gene expression normalized to wild type expression level = 1.0, which is indicated by the dashed line. *$P < 0.05$; **$P < 0.01$; ***$P < 0.001$ (unpaired *t*-test). **f** Representative immunoblots of $Ca^{2+}$-handling proteins. **g** Protein expression of transgenic animals compared to wildtype littermates obtained from immunoblots (6 months-old *mice* $n = 3$ animals/group). Two replicate immunoblots per animal, in pPln-S16 and pPln-T17 one replicate immunoblot per animal. Glyceraldehyde-3-phosphate dehydrogenase (GAPDH) was used as a loading control. *$P < 0.05$; **$P < 0.01$ (unpaired *t*-test). Data are shown as mean ± s.e.m. Source data are provided as a Source data file

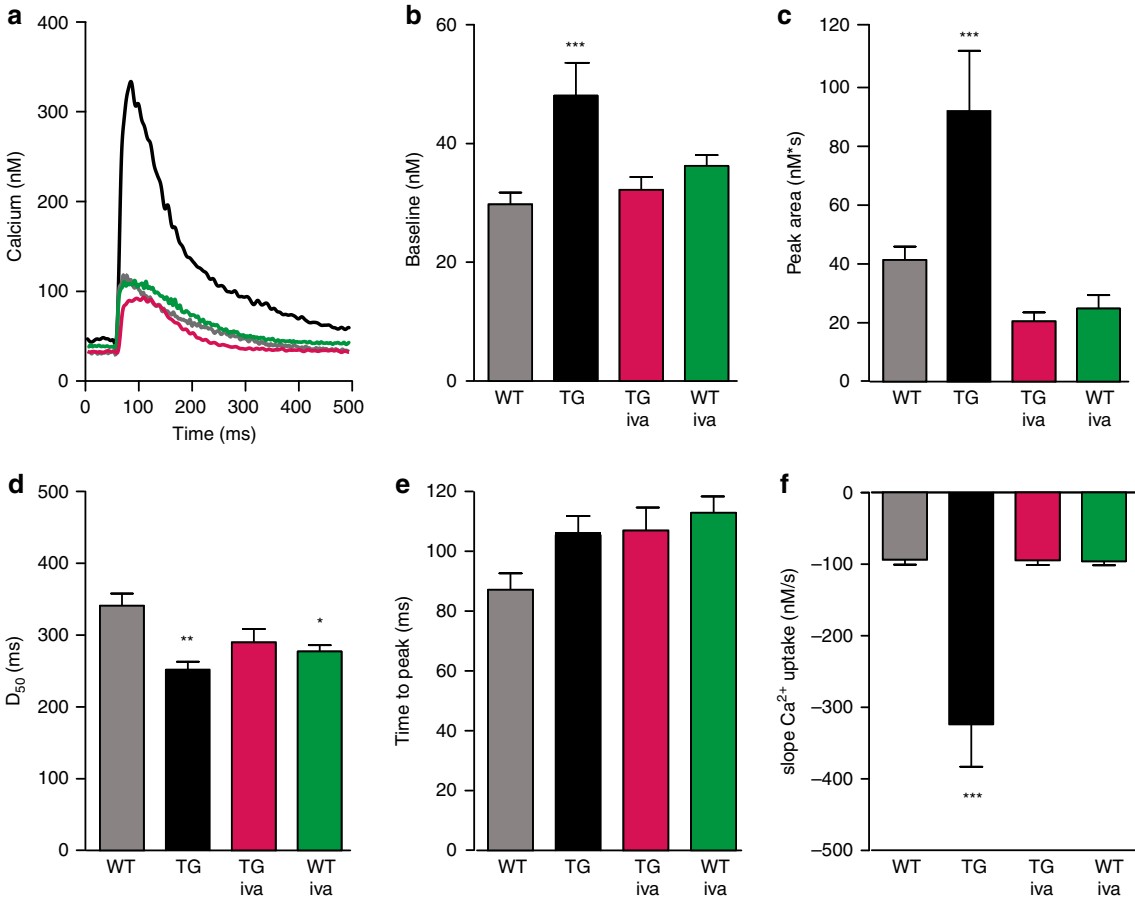

**Fig. 5** HCN4 overexpression affects intracellular $Ca^{2+}$ homeostasis. **a** Representative $[Ca^{2+}]_i$ transient traces measured as change in the Fluo-4 fluorescence in electrically driven cardiomyocytes from wild type [baseline (gray) and ivabradine treated (green)] and $HCN4^{tg/wt}$ [baseline (black) and ivabradine treated (magenta)] *mice* (3 months of age). **b**–**f** Quantitative analysis reveals a significant change of $[Ca^{2+}]_i$ to higher diastolic baseline levels (**b**), and larger amounts of total cellular $Ca^{2+}$ movement (reflected by the area under the curve) during the systole in $HCN4^{tg/wt}$ cardiomyocytes (**c**). Similar time to peak among groups indicates unaffected $Ca^{2+}$ release from sarcoplasmic reticulum (SR) by ryanodine receptor (RyR) activity (**d**). Quantification of time to half decay of $Ca^{2+}$ transients ($D_{50}$) (**e**), and slope $Ca^{2+}$ uptake (**f**), show that removal of increased baseline $[Ca^{2+}]_i$ to the SR, driven by SERCA, is augmented in $HCN4^{tg/wt}$ cardiomyocytes. Changes of $HCN4^{tg/wt}$ cardiomyocytes in $Ca^{2+}$ homeostasis were mostly reversed by treatment with the $I_f$ blocker ivabradine (3 μM) underlining that the overexpressed HCN4 current is the pivotal cause for the observed $Ca^{2+}$ imbalance. Data are expressed as mean ± s.e.m. (wild type: $n = 90$ cells, $HCN4^{tg/wt}$: $n = 42$ cells, iva-treated wild type: $n = 48$ cells, iva-treated $HCN4^{tg/wt}$: $n = 75$ cells derived from six animals/group; *$P < 0.05$, **$P < 0.01$, ***$P < 0.001$ compared to wild type; ANOVA followed by Tukey test). Source data are provided as a Source data file

repolarization were shorter in transgenic cells, with significant changes of $APD_{50}$ at all pacing frequencies and of $APD_{20}$ at 0.5 Hz. Evaluation of $APD_{90}$, by contrast, did not differ significantly between wild type and $HCN4^{tg/wt}$ cardiomyocytes (Fig. 8d–f).

**$HCN4^{tg/wt}$ cardiomyocytes are prone to arrhythmias.** Based on the overexpression of depolarizing $I_f$ and consecutive $Ca^{2+}$ overload we asked whether $HCN4^{tg/wt}$ cardiomyocytes elicit increased cellular automaticity and/or arrhythmogenesis.

Recordings of $[Ca^{2+}]_i$ transients and APs (Fig. 9a–g) frequently revealed sustained trains of automaticity in $HCN4^{tg/wt}$ cardiomyocytes (Fig. 9c), while such changes were only sparsely observed in wild type cells. Under steady state field stimulation the $[Ca^{2+}]_i$ transients of wild type cardiomyocytes quickly decayed (Fig. 9a), while a subset of $HCN4^{tg/wt}$ cardiomyocytes showed abnormal diastolic $[Ca^{2+}]_i$ clearance (Fig. 9b, Supplementary Fig. 2). Correspondingly, transgenic cardiomyocytes were prone to afterdepolarizations (ADs), which resulted in premature APs and arrhythmic firing (Fig. 9e, f). Among all cells

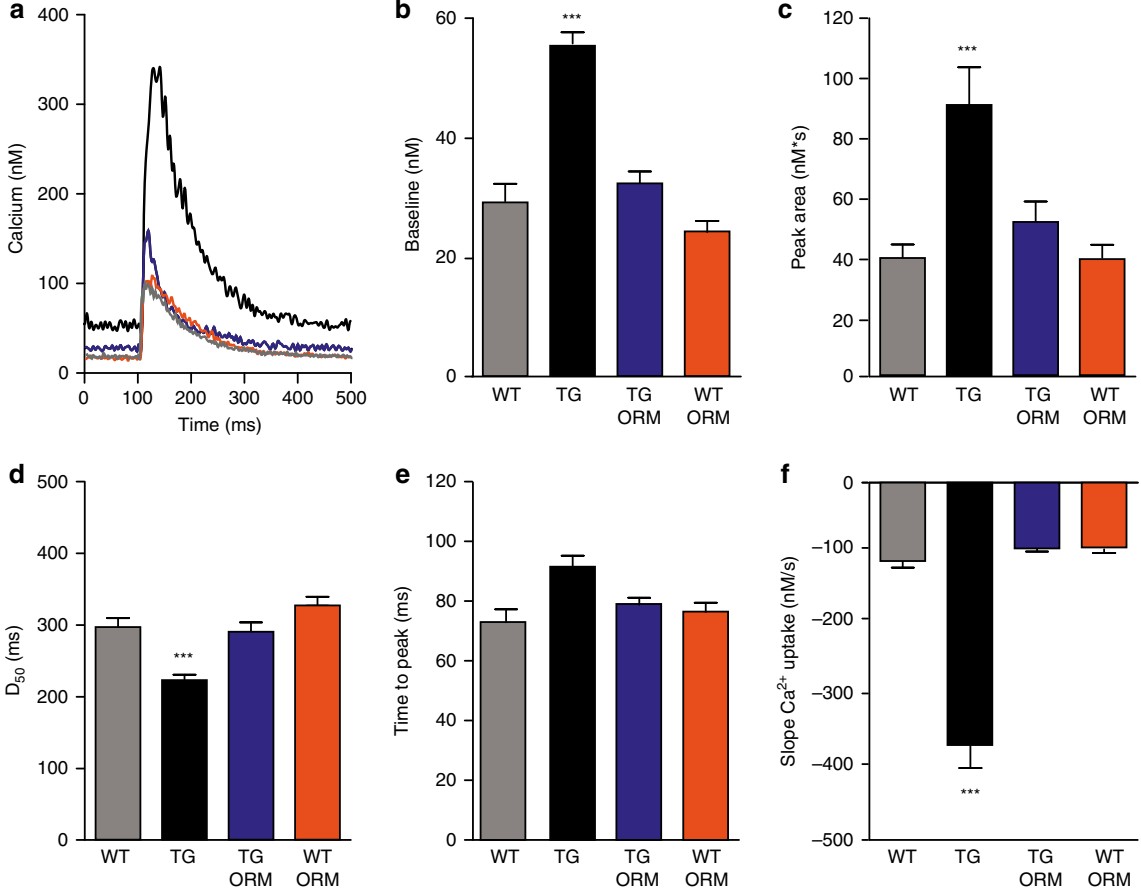

**Fig. 6** NCX blockade reverses HCN4-mediated changes of $Ca^{2+}$ homeostasis. **a** Representative $[Ca^{2+}]_i$ transient traces measured as change in the Fluo-4 fluorescence in electrically driven cardiomyocytes from wild type [baseline (gray) and ORM-10103-treated (orange)] and $HCN4^{tg/wt}$ [baseline (black) and ivabradine-treated (blue)] *mice* (3 months of age). **b**-**f** Quantitative analysis reveals changes in $Ca^{2+}$ homeostasis of $HCN4^{tg/wt}$ cardiomyocytes that were extensively recovered by treatment with ORM-10103. Diastolic $[Ca^{2+}]_i$, markedly increased in $HCN4^{tg/wt}$ cardiomyocytes, was significantly lower after ORM-10103 treatment (TG vs. TG ORM; ***$P < 0.001$; ANOVA) (**b**). Similarly, total cellular $Ca^{2+}$ movement (peak area) (TG vs. TG ORM; ***$P < 0.001$; ANOVA) (**c**), time to peak (TG vs. TG ORM; ***$P < 0.001$; ANOVA) (**d**) slope $Ca^{2+}$ uptake ((TG vs. TG ORM; ***$P < 0.001$; ANOVA) (**e**), and time to half decay ($D_{50}$) of calcium transients (TG vs. TG ORM; ***$P < 0.001$; ANOVA) (**f**), normalized to wild type levels after treatment of $HCN4^{tg/wt}$ cardiomyocytes with ORM-10103, indicating important implication of dysregulated NCX current in the changed $Ca^{2+}$ homeostasis of $HCN4^{tg/wt}$ cardiomyocytes. Data are expressed as mean ± s.e.m. (wild type: $n = 67$ cells, $HCN4^{tg/wt}$: $n = 102$ cells, ORM-treated wild type: $n = 72$ cells, ORM-treated $HCN4^{tg/wt}$: $n = 92$ cells derived from six animals/group; *$P < 0.05$, **$P < 0.01$, ***$P < 0.001$ compared to wild type; ANOVA followed by Tukey test). Source data are provided as a Source data file

recorded, the percentage of cells exhibiting ADs or virtually spontaneous firing, was significantly higher in transgenic (55.5 ± 7.6%; $n = 5$ hearts; 12–30 cells per heart) than in wild type cardiomyocytes (17.8 ± 3.4%; $n = 5$ hearts, 22–34 cells per heart; $P = 0.0002$; ANOVA followed by Tukey test) (Fig. 9g). To delineate the immediate contribution of $I_f$ to arrhythmogenesis within our model we treated the cells with 3 μM ivabradine (iva). Remarkably, iva treatment diminished spontaneous firing and ADs, and percentage of $HCN4^{tg/wt}$ cardiomyocytes showing arrhythmic behavior after treatment was not different from wild type (Fig. 9g), demonstrating an important contribution of increased $I_f$ to cellular arrhythmogenicity.

**Electrophysiological phenotype of HCN4$^{tg/wt}$ *mice*.** To evaluate ECG parameters and to seek for arrhythmias in $HCN4^{tg/wt}$ *mice* we performed ECG recordings under anesthesia and telemetric recordings in freely roaming animals. Both approaches revealed similar heart rates and conduction parameters among transgenic ($HCN4^{tg/wt}$) and wild type *mice* and did not exhibit significant changes of additional ECG parameters (Supplementary Fig. 4, Supplementary Table 1). However, increased numbers of

premature ventricular captures and few non-sustained ventricular tachycardias (mostly triplets) were recorded in transgenic but not in wild type animals (Fig. 9h, i), while sustained ventricular tachycardias were not observed. This indicates that $HCN4^{tg/wt}$ *mice* have a predisposition to ventricular arrhythmogenesis, although high-grade arrhythmias are not part of the phenotype.

## Discussion

We here report that augmentation of myocardial $I_f$ leads to the development of a cardiomyopathy phenotype with biventricular chamber dilation, significantly reduced wall thickness and decreased ejection fraction. To study the myocardial effect of increased $I_f$ under conditions that minimized its influence on heart rate, we have generated transgenic *mice* that express *human* HCN4 ($HCN4^{tg/wt}$) under control of the murine *cTNI* promoter. While abundantly expressed in the working myocardium, cTNI is downregulated in the SAN and the conduction system[22]. This is phenotypically reflected by telemetric recordings revealing similar heart rates and conduction parameters in transgenic and wild type *mice* (Supplementary Fig. 4, Supplementary Table 1), thus

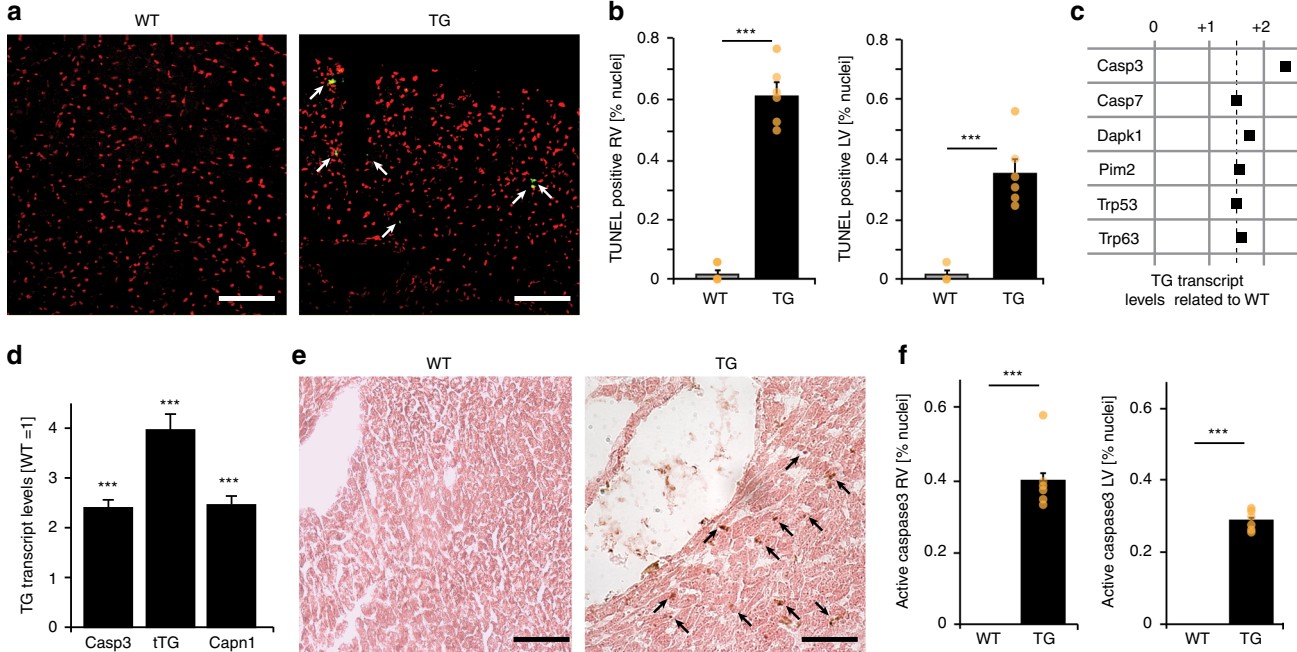

**Fig. 7** Assessment of myocardial apoptosis. **a** Representative TUNEL assay for detection of fragmented nuclear DNA (green) and nuclear counterstaining (red) in cryosections of right ventricles from wild type (left) and HCN4$^{tg/w}$t (right) mice (2 months of age). **b** Quantification of TUNEL-positive nuclei in right (RV) and left (LV) ventricular tissue from wild type and HCN4$^{tg/wt}$ mice revealed a significant increase in the number of cells with fragmented nuclear DNA in both ventricles of transgenic animals (n = 6 animals/group; ***P < 0.001; unpaired t-test). **c** Genes with transcriptional changes >1.5 fold in HCN4$^{tg/wt}$ compared to wild type hearts (values are normalized to corresponding wild type = 1.0) using the Mouse Apoptosis RT2 Profiler PCR Array (SABioscience). Pronounced transcriptional changes were observed in the apoptosis effector gene caspase-3 (for additional data, please refer to Supplementary Fig. 3). **d** qRT-PCR transcription analysis of the caspase 3 (Casp3), tissue transglutaminase (tTG), and calpain 1 (Capn1) genes in ventricles of HCN4$^{tg/wt}$ mice normalized to corresponding wild type littermates (wt = 1.0; n = 6 animals/group; ***P < 0.001; unpaired t-test). **e** Immunohistochemical detection of caspase-3 activity (brown) in representative sections of right ventricular tissue from wild type (left) and HCN4$^{tg/wt}$ (right) mice. **f** Quantitative analysis shows proportion of cells positive for active caspase-3 in right and left ventricular tissue from wild type and HCN4$^{tg/wt}$ mice (n = 6 animals/group; ***P < 0.001; unpaired t-test). Data are shown as mean ± s.e.m. Source data are provided as a Source data file

providing the opportunity to selectively study the consequences of HCN4 overexpression in the working myocardium.

There is growing evidence that ion channel disorders not only cause cardiac arrhythmias, but also contribute to structural abnormalities of the heart as well[23–26]. We and others reported[25,26] that HCN4 loss-of-function mutations are associated with noncompaction cardiomyopathy, pointing to an involvement of HCN4 in ventricular wall maturation at embryonic stages. Accordingly, HCN4 has been identified as primary cell marker for the cardiomyogenic progenitor pool of the first heart field, implicated in the earliest stage of heart formation[8,9]. During later development and adult stages, HCN4 is downregulated in the healthy myocardium, while abundant expression is restricted to the SAN and the conduction system[6]. Remarkably, in this context, HCN2 was recently shown to influence developmental brain morphology and function, as well[27].

It is well documented that I$_f$ and the expression of the main ventricular HCN isoforms 2 and 4 are significantly increased in ventricular moycytes of heart failure patients[5,11,28]. These changes are considered reminiscent of the immature myocytic phenotype, based on the fact that during pathological remodeling several early genes reappear in reverse order compared to embryonic development[29]. To mimic maladaptive upregulation of HCN4 in our model, the cTNI promoter was used to activate hHCN4 transcription from the intermediate-late phase of fetal development on[30], leading to augmented levels in the myocardium throughout postnatal and adult stages, normally characterized by low HCN4 expression[6]. At physiological RMP of ventricular cardiomyocytes (~−90 mV) patch clamp recordings

revealed a two–three-fold higher I$_f$ density in HCN4$^{tg/wt}$ cardiomyocytes compared to wild type (Fig. 1e, f), reflecting the magnitude of current increase that was found in cardiomyocytes of heart failure patients[5]. Accordingly, transgenic mice developed myocardial alterations within two months postpartum, supporting a primary maladaptive involvement of increased I$_f$. Likewise, we observed a rise of apoptosis and markers of the fetal gene program (Fig. 3a, b) with significantly upregulated cell growth markers e.g. glycogen synthase kinase-3 beta (GSK-3B) and mammalian target of rapamycin (mTOR) (Fig. 4b), indicating cardiac remodeling and activation of compensatory cellular responses. Thus, postnatal myocardium appears susceptible to augmented I$_f$ levels, and carefully regulated HCN patterning may constitute a prerequisite for the maintenance of structural cardiac integrity. In line, chronic peri- and postnatal treatment of transgenic mice with the I$_f$ channel blocker ivabradine antagonized the myocardial changes (Fig. 2), underlining that augmented I$_f$ is key to structural alterations occurring in HCN4$^{tg/wt}$ animals. Noteworthy, the heart rate lowering effect of ivabradine may contribute to protection from adverse remodelling as well.

Given the primary Na$^+$ conductance of HCN4 channels at cardiomyocyte RMP (~−90 mV)[31] and the close interrelation of intracellular Na$^+$ and Ca$^{2+}$ homeostasis[32], we asked how increased I$_f$ might influence Ca$^{2+}$ cycling. To elucidate the impact of increased I$_f$, we measured [Ca$^{2+}$]$_i$ transients in isolated, electrically driven cardiomyocytes in the pre- and absence of ivabradine. Strikingly, we observed a marked rise in [Ca$^{2+}$]$_i$ baseline levels and significantly augmented systolic [Ca$^{2+}$]$_i$ transients in transgenic cardiomyocytes – changes that were fully recovered in

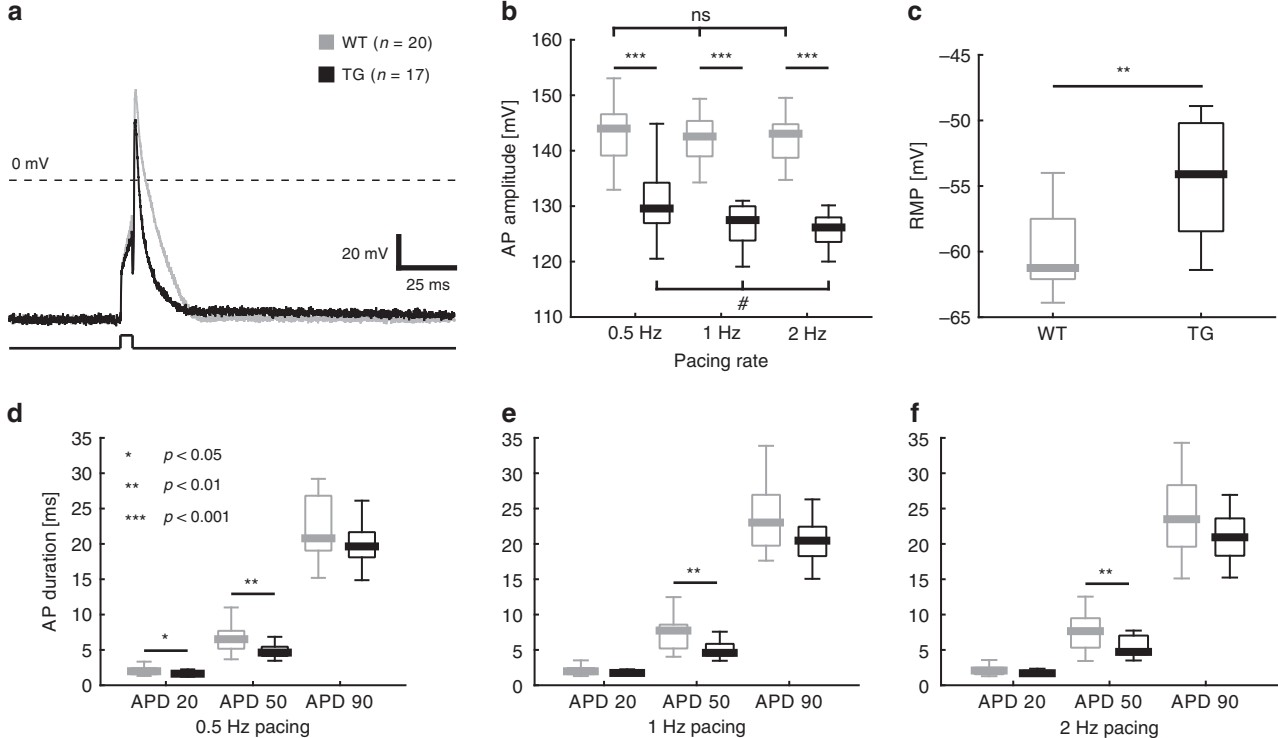

**Fig. 8** Electrophysiological properties of HCN4$^{tg/wt}$ cardiomyocytes. **a** Representative action potential (AP) traces of wild type (gray) and HCN4$^{tg/wt}$ cardiomyocytes (black). **b** Quantitative data on AP amplitude of wild type (gray) and HCN4$^{tg/wt}$ (black) cardiomyocytes at different pacing rates (0.5, 1, 2 Hz) showed decreased amplitude of overshoot in cells from transgenic *mice*. AP amplitudes of HCN4$^{tg/wt}$ cardiomyocytes decreased significantly with higher pacing rates. **c** HCN4$^{tg/wt}$ cardiomyocytes show depolarized RMP. **d–f** AP properties were evaluated at different pacing rates (0.5, 1, 2 Hz). AP duration at 20 (APD$_{20}$), 50 (APD$_{50}$), and 90% (APD$_{90}$) repolarization are depicted. Data are presented as boxplots. Box limits delineate interquartile values, centerlines represent the median and whiskers indicate the range (wild type: $n = 20$ cells, HCN4$^{tg/wt}$: $n = 17$ cells, for RMP: wild type: $n = 14$ cells, HCN4$^{tg/wt}$: $n = 12$ cells derived from $n = 5$ animals/group; 3 months of age; *$P < 0.05$, **$P < 0.01$, ***$P < 0.001$; unpaired t-test; for evaluation of rate-dependency of AP amplitudes: #$P < 0.05$; nonparametric ANOVA followed by Dunn's test). Source data are provided as a Source data file

the presence of ivabradine. Thus, in the myocardium chronic upregulation of I$_f$ exerts a diastolic Ca$^{2+}$ overload and importantly interferes with cellular Ca$^{2+}$ homeostasis.

As the primary mechanism of Ca$^{2+}$ efflux in cardiac myocytes is via electrogenic Na$^+$/Ca$^{2+}$ -exchange (NCX), Ca$^{2+}$ homeostasis is tightly linked to Na$^+$ regulation[32]. Under physiological conditions, NCX primarily operates in (forward) Na$^+$ -in/ Ca$^{2+}$ -out mode, and reduces diastolic [Ca$^{2+}$]$_i$ levels by extruding Ca$^{2+}$. In pathological states, however, when [Na$^+$]$_i$ is increased, NCX exerts 'reverse mode' function, moving Ca$^{2+}$ into the cell[32]. Notably, [Na$^+$]$_i$ is known to rise in a rate dependent fashion[32], which exacerbates [Ca$^{2+}$]$_i$ overload at high heart rates. The impact of I$_f$ on [Na$^+$]$_i$, in relation to membrane potential has been demonstrated in sheep Purkinje fibers by voltage-clamp recordings in early works[33,34] after the I$_f$ mixed Na$^+$ and K$^+$ ionic nature was originally described[35]. These studies showed that membrane hyperpolarization to below −60 mV significantly increased intracellular Na$^+$ activity, and related this increase to Na$^+$ influx through I$_f$, thus providing a direct link between [Na$^+$]$_i$ and the size of I$_f$. Based on these data, we hypothesized that HCN4 overexpression-mediated accumulation of [Na$^+$]$_i$ may raise Ca$^{2+}$ influx via reverse mode NCX activity (Fig. 10). According to this assumption, instant application of ORM-10103, an agent that effectively inhibits reverse mode NCX[20,36], abolished Ca$^{2+}$ overload, demonstrating the pathophysiological relevance of dysregulated NCX. Likewise, inhibition of increased Na$^+$ influx via HCN4 channels using ivabradine facilitated resetting of the NCX equilibrium towards the 'forward mode'

and restored [Ca$^{2+}$]$_i$ to wild type levels, similar to ORM-10103. Of note, depolarization of RMP by augmented I$_f$ may lower the driving force of forward NCX a fortiori, thereby aggravating chronic accumulation of [Ca$^{2+}$]$_i$ in HCN4$^{tg/wt}$ hearts. Based on these data it is reasonable to assume that inhibition of upregulated I$_f$ in clinical heart failure may attenuate Ca$^{2+}$ overload of the ventricular myocardium. Synergistically, heart rate lowering decreases [Na$^+$]$_i$[32] and therefore shifts the Na$^+$/Ca$^{2+}$ exchanger equilibrium towards the forward mode, which will alleviate [Ca$^{2+}$]$_i$ overload independently from ventricular I$_f$ blockade. Thus, besides improved energy supply through heart rate reduction the delineated mechanisms may importantly contribute to positive outcome of patients with heart failure and increased heart rates treated by I$_f$ blockade[13].

We next addressed the effects of increased I$_f$ and changed Ca$^{2+}$ homeostasis on cellular electrophysiology. Action potential recordings of isolated HCN4$^{tg/wt}$ cardiomyocytes showed depolarized RMP and reduced amplitude of overshoot in a rate-dependent fashion, consistent with HCN4-mediated diastolic Na$^+$ influx. Furthermore, APD$_{20}$ and APD$_{50}$ were shorter compared to WT, while APD$_{90}$ was similar. This in parts is consistent with previous data obtained from cardiomyocytes overexpressing HCN2[37], which showed that outward tail current of I$_f$ shortened APD at membrane potentials more positive than reversal potential (∼−35 mV for HCN4[38]), while inward tail current of I$_f$ prolonged APD at membrane potentials more negative than reversal potential. However, prolongation of APD$_{90}$ was not observed in our model, most likely due to the repolarizing driving

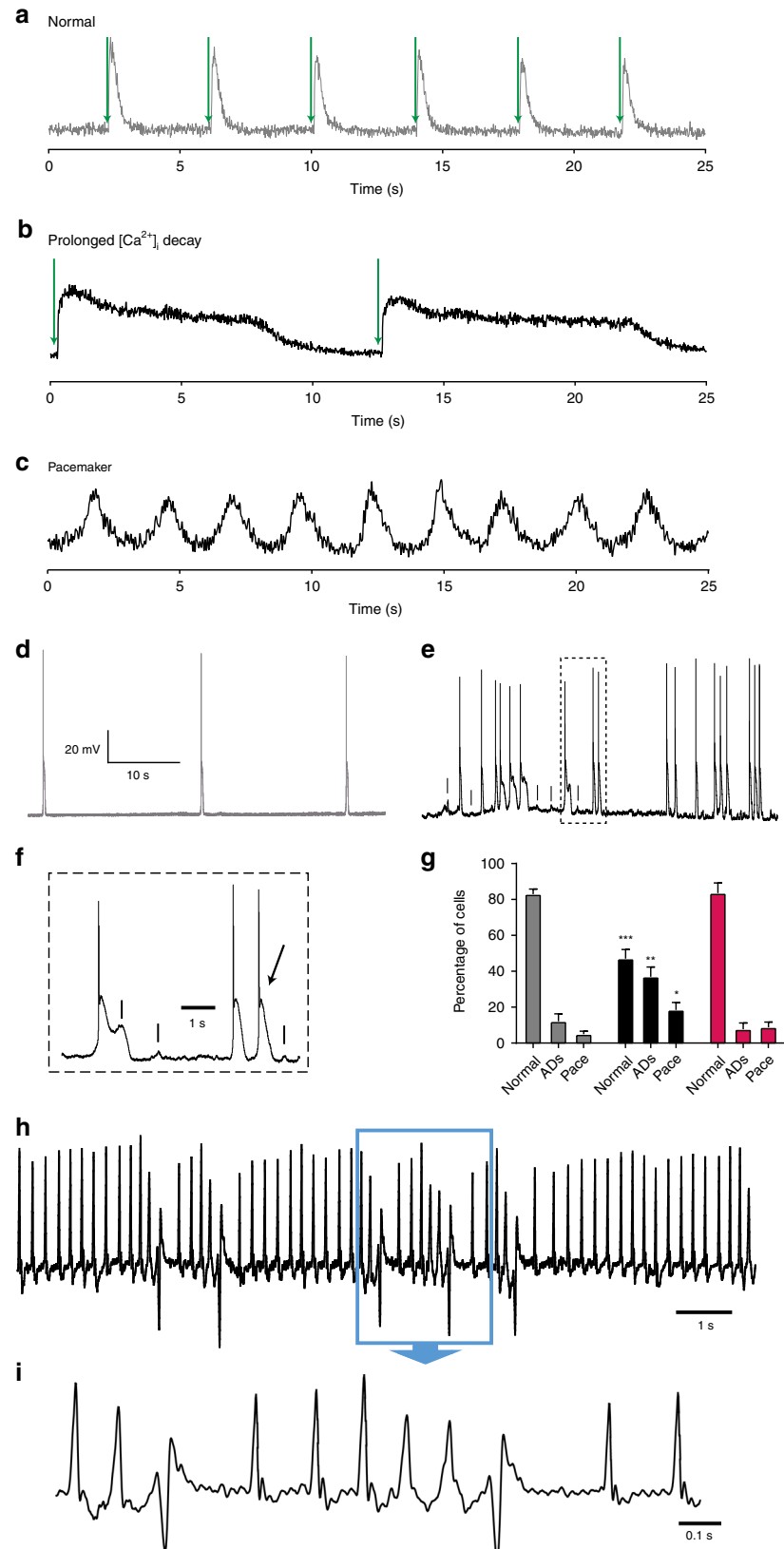

force of the electrogenic 'reverse mode' NCX, known to effectively shorten AP duration when $[Na^+]_i$ is increased[39].

Interestingly, we found that myocytes isolated from HCN4[tg/wt] hearts showed a higher incidence of spontaneous APs than wild type cells, and a subset of cells exerted virtual pacemaker function. The increased automaticity may originate from $I_f$ augmentation and not from spontaneous $Ca^{2+}$ cycling-mediated membrane depolarisation, as the NCX equilibrium is shifted toward 'reverse mode' leading to an outward, hyper-polarizing current. Moreover, HCN4[tg/wt] cardiomyocytes were

**Fig. 9** HCN4 overexpression leads to increased arrhythmogenicity. **a–c** Representative $[Ca^{2+}]_i$ transients measured as change in the Fluo-4 fluorescence in cardiomyocytes from *wild type* **a**, and *HCN4^{tg/wt}* **b**, **c** *mice* (3 months of age). **a** Normal behavior, exhibited by cardiomyocytes isolated from wild type *mice*, characterized by lack of automaticity (defined as spontaneous beating rate < 0.2 Hz) and rapid decay of $Ca^{2+}$ transients under field stimulation. In contrast, cardiomyocytes from *HCN4^{tg/wt}* mice frequently showed spontaneous pacemaker activity, characterized by periodic firing of $[Ca^{2+}]_i$ transients (**b**), or $[Ca^{2+}]_i$ transients with abnormal prolongation of diastolic $[Ca^{2+}]_i$ decay (defined as $D_{90} > 2000ms$) (**c**). Field stimulation in **a**, **b** is denoted by green arrows. **d–f** Recording of spontaneous action potentials (APs) in wild type (**d**), and HCN4^{tg/wt} (**e**, **f**) cardiomyocytes. While most wild type cells lacked relevant automaticity (defined as spontaneous beating rate < 0.2 Hz) (**d**), a subset of *HCN4^{tg/wt}* cells showed spontaneous pacemaker activity, with or without afterdepolarizations (ADs). ADs (denoted by dash in **e**, **f**) frequently induced spontaneous APs (denoted by arrow in **f**), leading to arrhythmic firing. **f** depicts a detail of **e**. **g** Quantitative analysis of spontaneous activity of wild type (gray), *HCN4^{tg/wt}* (black) and ivabradine-treated *HCN4^{tg/wt}* (magenta) cardiomyocytes indicate a greater proportion of cells with pacemaker activity (Pace) or ADs in *HCN4^{tg/wt}* cardiomyocytes compared to wild type. Remarkably, spontaneous firing and ADs were diminished after treatment with ivabradine, and percentage of cells showing arrhythmic behavior was not different from wild type thereafter. Data are expressed as mean ± s.e.m. ($n = 5$ independent experiments [*mice*] per group, 12–34 cells evaluated per experiment; *$P < 0.05$, **$P < 0.01$, ***$P < 0.001$ compared to wild type; ANOVA followed by Tukey test). **h**, **i** Telemetry recording from a *HCN4^{tg/wt}* mouse showing an episode with premature ventricular captures that occur singular and as triplet. **i** depicts a detail of **h**. Source data are provided as a Source data file

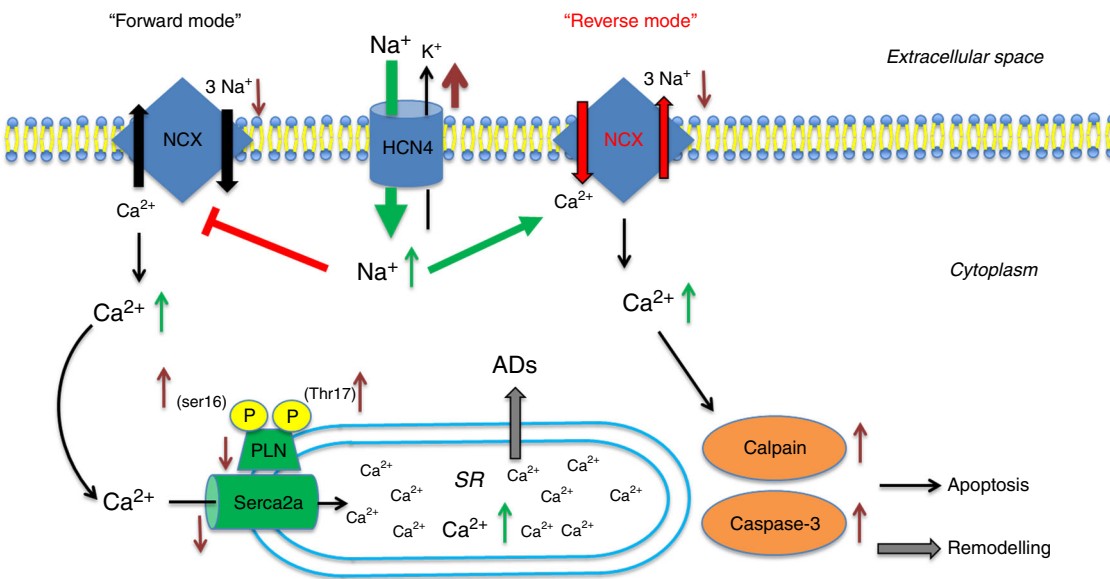

**Fig. 10** Proposed pathogenetic processes. Model of how augmented HCN4 expression in the myocardium may contribute to pathological remodeling. HCN4 upregulation mediates diastolic $Na^+$ influx into the cell, leading to a rise of $[Na^+]_i$. $Ca^{2+}$ homeostasis is tightly linked to $Na^+$ regulation via the NCX activity. Under physiological conditions NCX primarily operates in (forward) $Na^+$-in/ $Ca^{2+}$-out mode, reducing diastolic $[Ca^{2+}]_i$. High $[Na^+]_i$ gives rise to a shift of the NCX equilibrium towards 'reverse mode' thereby leading to increased $[Ca^{2+}]_i$. Changed cytoplasmic $Ca^{2+}$ cycling, in turn, interferes with SR calcium uptake resulting in increased SR $Ca^{2+}$ stores, driven by augmented phosphorylation of PLN. Sarcoplasmic $Ca^{2+}$ overload causes afterdepolarizations (ADs) and cardiac arrhythmogenesis, and high diastolic $[Ca^{2+}]_i$ activates μ-calpain $Ca^{2+}$-sensor- and caspase-3-related apoptosis leading to adverse remodeling. Brown arrows indicate regulation of protein expression; green arrows denote changes of ion concentration

prone to afterdepolarizations resulting in trains of premature APs. Recording of calcium transients showed abnormal diastolic $Ca^{2+}$ decay in 39% of HCN4^{tg/wt} cells, produced by SR $Ca^{2+}$ overload, promoting afterdepolarizations and arrhythmic firing[40]. However, during telemetric recordings HCN4^{tg/wt} animals exhibited no sustained arrhythmias, although higher numbers of PVCs and few nonsustained VTs indicated increased arrhythmogenicity compared to wild type animals. This imbalance between cellular arrhythmogenicity and lack of high-grade arrhythmias in vivo might be explained by source-sink mismatch[41] proposing that propagation of ADs originating from single cells (source) is hindered by the rectifying potassium conductance of neighboring myocytes within the myocardial syncytium (sink), thereby protecting the heart from deleterious arrhythmias.

One well-known hallmark of failing heart cells is disturbances in calcium cycling[42–44]. Harmful $[Ca^{2+}]_i$ promotes cardiomyocyte death through apoptotic or necrotic pathways[43]. Previous studies have delineated the association between apoptotic cell-loss and progression of heart failure in animal models and in *humans*[45,46]. Moreover, $Ca^{2+}$ overload induced by persistently activated L-type calcium channels[40,43,47] or chronically compromised $Ca^{2+}$ removal systems[47] were shown to induce myocardial cell death and remodeling. Accordingly, the structural changes in our model can be explained by the observed caspase-3-mediated apoptosis. Likewise, we found higher transcript levels of tissue transglutaminase and the $Ca^{2+}$-sensor protease μ-calpain (Fig. 3d), implicated in the activation of myocardial apoptosis and structural remodeling in association with $[Ca^{2+}]_i$ overload[21]. Basically, upregulation of $I_f$ in heart failure might be seen as cellular short term mechanism that increases SR calcium load to provide more inotropic support. However, this is at the price of chronic $[Ca^{2+}]_i$ accumulation resulting in cell death and disease progression.

In summary, we show that a two–three-fold increase of $I_f$ in cardiac myocytes, which is comparable to levels observed in *human* heart failure, affects cardiac structure and promotes cellular arrhythmogenicity in HCN4$^{tg/wt}$ *mice*. Activation of reverse-mode NCX by increased HCN4-mediated Na$^+$ influx leads to augmented $[Ca^{2+}]_i$ and dysregulated $Ca^{2+}$ homeostasis driving apoptosis and adverse cardiac remodeling. This might have particular implications in disease states associated with increased $I_f$, affecting calcium cycling as well as excitation-contraction coupling. Thus, direct cardioprotective mechanisms in addition to improved energy supply at lower heart rates may underlie the beneficial effects of $I_f$ inhibition in heart failure. Our findings provide insight in previously undescribed cardiac pathomechanisms, bridging a changed electrical state to a structural phenotype, which importantly could influence future treatment strategy of heart failure.

## Methods

**Generation of transgenic mice**. Transgenic *mice* were generated in a C57Bl/6NCrl background (Charles River Laboratories, Wilmington, MA, USA) by pronuclear injection of an 8.2 kb DNA fragment carrying a 4.3 kb promoter fragment of the murine *cTnI* gene[18,19] joined to the 5′ end of the *human HCN4* cDNA[48] that was fused to the bovine growth hormone gene poly A signal (Fig. 1a). Tail DNA analysis revealed four different founders that were subsequently bred with C57BL/6 N wild type *mice* to generate progeny that appeared healthy and fertile.

**Care and use of mice**. Mixed genotype groups of each gender of no more than five animals were housed in standard *mouse* cages under specific pathogen-free conditions (12:12-h dark–light cycle, constant temperature, constant humidity and food and water ad libitum). All experiments were carried out in accordance with the Guide for the Care and Use of Laboratory Animals published by the US National Institute of Health (NIH publication number 85–23, revised 1996), and with the European Community guidelines for the use of experimental animals as well as all relevant regulations. Protocols were approved by the local regulatory authority (#35-9185.81/G-20/11 and #35-9185.81/G-226/16 Regierungspräsidium Karlsruhe, Germany). We used male *mice* for all experiments.

**Surface and telemetric ECG recordings**. For surface ECG recordings *mice* were anesthetized with isoflurane vapor titration to maintain the lightest anesthesia possible[49]. On average, 1.5% vol/vol isoflurane was required to maintain adequate anesthesia. *Mice* were placed on a heating pad with continuous monitoring of body temperature (via a rectal probe) maintaining at 37 °C. Surface ECGs were recorded using clamp electrodes attached to each limb and to the chest in a midsternal position. The electrodes were connected to a Powerlab System (AD Instruments, Hastings, UK) and ECG recording were performed with a high and low frequency cut off of 1 kHz and 0,1 kHz. ECG recordings were analysed by an investigator blinded to the experimental groups using the Chart 5 software (AD Instruments, Hastings, UK). In addition to average heart rate, at least 30 complexes of each trace were signal averaged and the following parameters were evaluated in lead II: PR-interval, QRS complex duration, and QT-interval. Corrected QT intervals were calculated according to Mitchell et al., 1998[50], QTc = QT/(RR/100)1/2.

For telemetric ECG recordings, a radiotelemetry transmitter (model EA-F20, Data Sciences International, St. Paul, Minnesota, USA) was surgically inserted into the peritoneal cavity. *Mice* were anesthetized intraperitoneally with a mixture of ketamine (100 mg/kg) and xylazine (3 mg/kg). The leads were tunnelled under the skin to the recording sites, i.e. one subcutaneous lead placed at the right shoulder and a second subcutaneous lead placed at the lower left thorax. The abdominal skin was closed and *mice* were kept on a heating pad (37–38 °C) until they were fully recovered from anaesthesia. Before measurement, *mice* were allowed to recover for 1 week, followed by 24 h recording. ECG signals were converted to digital output and stored on a computer using the Powerlab System (AD Instruments, Hastings, UK), analysis was carried out with the Chart 5 software (AD Instruments, Hastings, UK).

**Transthoracic echocardiography**. The animals were sedated with low-dose isoflurane (1,5% vol/vol) and placed on a heating pad in supine position. A rectal probe was inserted to monitor the body temperature to maintain it at 37 °C. Chest hair was carefully removed with depilatory cream. Warmed echo gel was placed on the animal's chest. The Vevo® 2100 device (VisualSonics, Toronto, Canada) with the corresponding transducer MS-400 was used to record the echocardiographies. The investigator was blinded with respect to the treatment group. *Mice* were shaved, and left ventricular parasternal short-axis views were obtained in M-mode imaging at the papillary muscle level. Three consecutive beats were used for measurements of left ventricular end-diastolic internal diameter (LVEDD) and

left ventricular end-systolic internal diameter (LVESD). Fractional shortening (FS) was calculated as FS% = ((LVEDD − LVESD)/LVEDD) × 100.

**Transcription analysis**. Transcription was assayed by amplification of reverse transcribed total RNA isolated from whole ventricle tissue of wild type and transgenic *mice*. Hearts were excised, ventricular tissue was carefully dissected and immediately frozen in liquid nitrogen. Total RNA was extracted using TRIzol-Reagent® (Invitrogen, Karlsruhe, Germany) according to the manufacturer's instructions, followed by reverse transcription (Superscript III, Invitrogen, Karlsruhe, Germany). Quantitative real-time PCR (RT-qPCR) was performed using an ABS 7500 Realtime PCR System (Thermo Fisher, Waltham, MA, USA) according to the manufacturer's protocol. 96 well optical detection plates (Thermo Fisher, Waltham, MA, USA) were loaded to a total volume of 10 μl per well, consisting of 0.5 μl cDNA, 5 μl TaqMan Fast Universal Master Mix (Thermo Fisher, Waltham, MA, USA), and 6-carboxyfluorescein (FAM)-labeled pre-designed TaqMan primers (TaqMan Gene Expression Assays, Thermo Fisher, Waltham, MA, USA) specific to *human* and *mouse HCN4* transcripts, HCN1, HCN2. qRT-PCR of apoptosis and $Ca^{2+}$ processing related genes was performed using TaqMan assays for *Casp3, tTG, Capn1, RyR2, Serca2a, Pln, Na$^+$/K$^+$-ATPase* and *NCX1*. Ion channel transcripts were examined using TaqMan assays for *SCN5A, Ca$_v$1.2, Ca$_v$1.3, Ca$_v$2.1, Ca$_v$3.1, K$_{ir}$2.1, K$_v$1.4, K$_v$1.5, K$_v$4.2, K$_v$7.1, hERG,* and *KChip2*. Transcripts of myocardial cell growth marker genes and genes of the fetal gene program were determined with TaqMan assays for *mTOR, GSK-3B, ANP, Calcineurin* and *β-MHC*. Predesigned primers and probes detecting the housekeeping genes glyceraldehyde 3-phosphate dehydrogenase (GAPDH), hypoxanthine-guanine phosphoribosyltransferase 1 (HPRT1), and beta-actin (ACTB) were applied and normalization was carried out by a modified threshold cycle (CT) relative quantification method using GAPDH (Figs. 1b, 4a, 7d) or all three housekeeping genes (Fig. 4b–e), as published elsewhere[51]. All TaqMan primers used are listed in Supplementary Table 2. RT-qPCR reactions were performed in triplicate, and data are expressed as an average of triplicates.

Expression of apoptosis related genes was screened with Mouse Apoptosis RT² Profiler PCR Array (PAMM-012A-2, SABiosciences, Frederick, MD, USA) on an ABS 7500 Realtime PCR System and data was analysed using online software "RT² Profiler PCR Array Data Analysis" (version 3.5) on the manufacturer's website http://pcrdataanalysis.sabiosciences.com/pcr/arrayanalysis.php.

**Protein analysis**. Protein immunodetection was performed by sodium dodecyl sulfate (SDS) gel electrophoresis. Tissue sections obtained from indicated *mouse* left and right ventricle samples were rinsed in phosphate buffered saline (PBS), rapidly frozen in liquid nitrogen and stored at −80 °C. Tissue samples were homogenized (TissueRuptor, QIAGEN, Hilden, Germany) in a radio-immunoprecipitation (RIPA) lysis buffer containing 50 mM Tris-HCl (pH 7.4), 0.5% NP-40, 0.25% sodium deoxycholate, 150 mM NaCl, 1 mM EDTA, 1 mM Na$_3$VO$_4$, 1 mM NaF, and protease inhibitors (Complete; Roche, Indianapolis, IN, USA). The protein concentration was determined using the bicinchoninic acid (BCA) protein assay (Thermo Scientific, Rockford, IL, USA), and equal amounts of protein were separated on SDS polyacrylamide gels. Nitrocellulose membranes were developed by sequential exposure to blocking reagent containing 3% bovine serum albumin and 5% dry milk. Primary antibodies were directed against HCN4 (1:200 diluted in 5% dry milk), NCX1 (1:250), Cav1.2 (1:500), Serca2 (1:1000), Phospholamban (1:500), phosphorylated Phospholamban Thr-17 (1:2000), phosphorylated Phospholamban Ser-16 (1:2000) and Calpain1 (1:2000). Appropriate HRP-conjugated secondary antibodies (anti-*rabbit*: 1:500, anti-*mouse*: 1:10,000) were used. Signals were developed using an enhanced chemiluminescence assay (ECL Western Blotting Reagents, GE Healthcare, Buckinghamshire, UK) and quantified with ImageJ 1.41 Software (National Institutes of Health, Bethesda, MD, USA). Protein content was normalized to glyceraldehyde 3-phosphate dehydrogenase (GAPDH) using anti-GAPDH primary antibodies (1:40,000) and corresponding secondary antibodies for digital quantification of optical density. For detailed information on primary and secondary antibodies used please refer to Supplementary Table 3. Western blot images used to compose result panels are available in Supplementary Fig. 5.

**Isolation of cardiomyocytes**. Cardiomyocytes for intracellular calcium measurements were obtained following the Liao & Jain protocol[52]. Briefly, the *mice* received 200 IU heparin i.p. prior to sacrifice. The thoracic chamber was opened and a cannula with perfusion solution was inserted from the atria. The heart was harvested and perfused in the Langendorff system with Perfusion Buffer (in mM: 135 NaCl, 4 KCl, 1 MgCl$_2$, 10 HEPES, 0.33 NaH$_2$PO$_4$, 10 glucose, 20 2,3-butanediones monoxime, 5 taurine, pH7.2 at 37 °C for 5 min), followed by Digestion Buffer treatment (0.3 mg/g body weight collagenase D (Roche, Indianapolis, IN, USA), 0.4 mg/g body weight collagenase B (Roche, Indianapolis, IN, USA), 0.05 mg/g body weight protease XIV (Sigma Aldrich, St. Louis, MO, USA) in 25 ml Perfusion Buffer) until the heart muscle was pale and some signs of extracellular matrix dissociation and for additional 5 min with Perfusion Buffer to stop the dissociation. The cardiomyocytes were mechanically dissociated in Transfer Buffer A in mM: 135 NaCl, 4 KCl, 1 MgCl$_2$, 10 HEPES, 0.33 NaH$_2$PO$_4$, 5.5 glucose, 10 2,3-butanediones monoxime, 5 mg/ml bovine serum

albumin (Sigma Aldrich, St. Louis, MO, USA), pH 7.4 at 37 °C) and then plated on ECM-coated (Sigma Aldrich, St. Louis, MO, USA) petri dish (Zell-Kontakt, Nörten-Hardenberg. Germany). The extracellular calcium concentration was increased gradually in three consecutive steps from 0 mM (in Transfer Buffer A) to 0.06, 0.24 and 1.2 mM every 5 min. The cells where then incubated at 37 °C in 5% $CO_2$.

**Cellular electrophysiology.** $I_f$ was recorded under voltage-clamp conditions (whole-cell configuration) in isolated *mouse* ventricular cardiomyocytes at room temperature (21–23 °C), essentially as published[53,54]. In detail, patch-clamp measurements of *mouse* ventricular cardiomyocytes were performed in a solution containing (in mM): 137 NaCl, 25 KCl, 8 BaCl, 2 CaCl₂, 2 MnCl, 0.2 CdCl, 0.5 4-Aminopyridine, 5 HEPES, 10 Glucose, pH corrected to 7.4 using 1 M NaOH. The patch pipettes pulled from borosilicate glass (GB-150-8P, Science Products, Hofheim Germany) were generated on a DMZ Universal puller (Zeitz Instruments, Munich, Germany) and fire polished to give a final resistance of 2.5–4 MΩ. The pipette solution contained (in mM): 130 K-Aspartate, 5 Na₂ATP, 5 CaCl₂, 2 MgCl₂, 11 EGTA and 10 HEPES, pH corrected to 7.3 using 1 M KOH. Pharmacological agents were added to the bath solution to block L- type calcium currents (10 μM nisoldipine), the slow component of delayed rectifier K⁺ current ($I_{Ks}$, 10 μM chromanol), the rapid component of delayed rectifier K⁺ current ($I_{Kr}$, 10 μM E4031), ATP-dependent K⁺ current ($I_{KATP}$, 1 μM glibenclamid) and inward rectifier K⁺ currents ($I_{K1}$, 50 μM BaCl₂). $I_f$ currents were normalized to cell capacitance which was obtained from the time constant of a current transient evoked by a 5 mV potential step at the beginning of each sweep.

Recordings of cardiac action potentials were carried out in an extracellular solution containing (in mM): 137 NaCl, 5.4 KCl, 1.8 CaCl₂, 0.5 MgCl₂, 10 HEPES, 5.5 glucose, and a pipiette solution containing (in mM): 120 K-Aspartate, 10 KCl, 5 NaCl, 2 MgCl₂, 10 HEPES. For quantification of AP duration and amplitude, cells were held in current-clamp mode and current was injected to achieve a membrane potential of −90 mV. If injected currents exceeded 200pA, recordings were discarded and not included in analysis. APs were evoked by 5 ms suprathreshold depolarizing current injection at 0.5, 1, and 2 Hz frequencies. Quantification of resting membrane potential, and recordings of spontaneous APs and afterdepolarizations were performed without current injection.

$I_f$ and AP recordings were done with an Axopatch 200B amplifier (Molecular Devices, Sunnyvale, USA), digitized at 20 kHz with a 1401 Power3 Analog/Digital Converter (CED, Cambridge, UK) stored on a hard drive, and analysed off-line with custom MATLAB routines (Mathworks, Natick, USA) software. Recordings with less than 10% leak current were considered for data analysis. No leak subtraction was performed during the experiments. Recordings were discarded if access resistance exceeded 20 MOhm or changed for more than 20% during the recording period.

**Recording of calcium transients.** The cardiomyocytes were loaded with 10 μM Fluo-4, AM (Molecular Probes, Leiden, The Netherlands) dissolved in Transfer Buffer B (in mM: 137 NaCl, 5.4 KCl, 1.8 CaCl₂, 0.5 MgCl₂, 10 HEPES, 5.5 glucose, pH 7.4 at 37 °C) for 30 min. For pharmacological treatments ivabradine (Molekula, Munich, Germany; cat. no. 89982651/155974-00-8) or ORM-10103 (Sigma-Aldrich, Germany; cat. no. SML0972) were dissolved in DMSO for stock solution. For working solution, both ivabradine or ORM-10103 were dissolved in Transfer Buffer B at 3 μM or at 10 μM, respectively, and incubated for 10 min prior to the $Ca^{2+}$ recordings.

$[Ca^{2+}]_i$ transients were recorded using an Olympus OSP-3 System fluorescence microscope. Cardiomyocytes were single twitched electrically stimulated at 25 V for 10 ms at constant rate of 0.2 Hz. The fluorescence signal was obtained from cytosolic area and calibrated by a pinhole of 7.5 μm diameter, integrated in the photomultiplier, and A/D converted using PowerLab 4/35 and LabChart V7.0.

Out of 8–10 $Ca^{2+}$ transients recorded, the least five were analyzed, avoiding the non-physiological $Ca^{2+}$ transients due to time with no electrical stimulation between cell recordings. $Ca^{2+}$ transients were eligible for regular analysis when lacking automaticity and showing regular $[Ca^{2+}]_i$ clearance under field stimulation at 0.2 Hz. Cells that exhibited pacemaker activity (defined as spontaneous beating rate < 0.2 Hz) or abnormal diastolic $[Ca^{2+}]_i$ decay (defined as $D_{90} > 2000$ms) were excluded from regular analysis and evaluated in a separate record (please refer to Supplementary Fig. 2).

Calibration was performed at the end of each experiment in freshly loaded cells with Fluo-4, AM to transform voltage values into $[Ca^{2+}]_i$[55]. Briefly, cells were treated with high $Ca^{2+}$ solution (in mM: 140 NaCl, 5 KCl, 1.2 KH₂PO₄, 1.2 MgCl₂, 4 CaCl₂, 20 HEPES, 0.005 ionomycin, 0.01 CPA, 5 caffeine, 1 ouabain) to obtain Fmax values followed by application of zero-$Ca^{2+}$ solution (in mM: 140 LiCl, 5 KCl, 1.2 KH₂PO₄, 1.2 MgCl₂, 20 HEPES, 4 EGTA, 0.005 ionomycin, 0.01 CPA, 5 caffeine, 1 ouabaine) to provide Fmin value. The $Ca^{2+}$ signal (F) was then converted into $[Ca^{2+}]_i$ according to Grynkiewicz et al. 1985[56].

$[Ca^{2+}] = Kd(F\text{-Fmin})/(Fmax\text{-}F)$, with Kd: apparent $Ca^{2+}$ dissociation constant of the Fluo-4 of 345 nM, according to the manufacturer's information.

Five biophysical parameters were analyzed: baseline (μM), peak area (area under the curve; μM*s), $D_{50}$ (duration of the transient at 50% of the peak; ms), time to peak (time required to reach the maximum from the baseline; ms), slope $Ca^{2+}$

uptake time (time required to return to the base line from the peak to 5% of the baseline; nM/s).

**Peri- and postpartal in vivo treatment with ivabradine.** For treatment with the $I_f$ blocker ivabradine (Molekula, Munich, Germany) timed-pregnant $HCN4^{tg/wt}$ and wild type females were subcutaneously implanted with an osmotic mini pump (Alzet, Cupertino, CA, USA) at 18.5 days post-gestation. Ivabradine was delivered at a rate of 1.5 mg/kg/d for 3 weeks to the lactating female, and individual pumps were implanted in the pups from day 20 until 2 months postpartum.

**Histochemical analysis and apoptosis assays.** Hearts were dissected from transgenic and control *mice* at the age of 2 months, immediately frozen in isopentane on dry ice and stored at −80 °C. Serial coronal sections of 10 μm were produced for histochemical and immunohistochemical analyses.

For antibody staining, sections were fixed in 10% formaldehyde/PBS pH 7.2 (15 min), washed in PBS (3 × 5 min), permeabilized in 0.5% TritonX/PBS (30 min), blocked in 5% BSA/PBS and exposed for 24–48 h to the primary antibody in blocking solution at 4 °C. Primary antibodies used: rabbit polyclonal anti-HCN4 antibody (1:200; APC-052; Alomone labs, Jerusalem Israel); purified *mouse* monoclonal anti-caveolin 3 antibody (1:500; 610421; BD Transduction Laboratories, Palo Alto, CA, USA); anti-active caspase-3 (1:10; ab2302; Abcam, Cambridge, UK). Sections were washed in PBS (3 × 5 min) and incubated for 1 h at room temperature with a fluorescent labelled secondary antibody (1:500; MFP555 Goat anti-Rabbit IgG, MFP 488 goat anti-*mouse* IgG; MoBiTec, Goettingen, Germany) or anti-rabbit HRP-conjugated secondary antibody (Vectashield ABC Kit, Vector Labs, Burlingame, CA, USA). Sections were counterstained for 5 min with propidium iodide (1:500; ab14083; Abcam, Cambridge, UK) at room temperature.

Hematoxylin-eosin and Masson's trichrome stainings were performed according to standard protocols. In detail, cardiac preparations were fixed in 4% formaldehyde, embedded in paraffin, cut to 6 μm thickness and mounted onto SuperFrost Plus slides (Gerhard Menzel GmbH, Braunschweig, Germany). Sections were then stained with hematoxylin-eosin to analyse cardiac histology or Masson's trichrome dye to identify interstitial fibrosis. Blue staining following Masson's trichrome application indicates fibrotic tissue. Cardiac fibrosis was quantified at 10-fold magnification. The extent of fibrosis was defined as the ratio between fibrotic regions and total tissue area [% fibrosis = (area exhibiting fibrosis/total area)*100]. The calculation was based on averaged measurements from 10 sections per animal (right ventricle, five; left ventricle, five). Bright-field images were digitally recorded on Axioplan2 microscope (Zeiss, Oberkochen, Germany) with attached 3CCD camera (Intas, Goettingen, Germany).

TUNEL assay was performed and quantified according to manufacturer's protocol (DeadEnd™ Fluorometric TUNEL System, Promega). Confocal images of red and green immunofluorescence staining of heart tissue sections were acquired on a Leica confocal laser scanning unit TCS NT, which was coupled to a Leica DM IRB microscope. Acquisition of image series was performed using TCS NT (Leica, Heidelberg, Germany) software. All images were processed in ImageJ version 1.47 (NIH, Bethesda, MD, USA) software using standard quantification methods and standard ImageJ plug-ins.

**Electron microscopy.** Right and left ventricular wall tissue of $HCN4^{tg/wt}$ and wild type *mice* was dissected, and prepared for microscopy[57]. Briefly, cardiac tissue was fixed overnight at 4 °C in a calcium-free buffer (100 mM PIPES, pH 7.4 with 1.25% glutaraldehyde and 2% paraformaldehyde). After several washes with 100 mM PIPES, pH 7.0, samples were incubated for 2 h at room temperature in aqueous 2% osmium tetroxide and 1.5% potassium-hexacyanoferrate solution, washed in water for 3 h and incubated in water with 4% uranyl acetate overnight at 4 °C. Samples were dehydrated as described[57] and embedded in Epon medium. Ultra-sections were contrasted in uranyl acetate for up to 20 min and in lead citrate for 5 min. Sections were viewed on Philips EM400T (Philips IS, Eindhoven, The Netherlands) and images were documented on Image Plates and scanned with the Micro Imaging Plate Scanner (DITABIS, Pforzheim, Germany).

**Statistical analysis.** All experiments and primary analyses were blinded. Statistical analysis was performed using GraphPad Prism 5.0 (GraphPad Prism Inc. La Jolla, CA). Statistical significance and comparisons between samples were analyzed by a two-tailed unpaired Student's $t$-test. Comparisons between multiple groups were performed using one-way ANOVA followed by Bonferroni (for calcium recordings), Tukey (for histochemical analysis) or Dunn's (for action potential amplitudes) post-hoc tests. Differences were considered significant at a level $P < 0.05$. Data are presented as arithmetic mean ± s.e.m. or median ± maximum/minimum.

**Reporting summary.** Further information on research design is available in the Nature Research Reporting Summary linked to this article.

## Data availability

The data that support the findings of this study are available from the article and the Supplementary Information Files, or from the corresponding author upon reasonable

request. A reporting summary for this article is available as a Supplementary Information Files. Source data underlying the Figs. 1b and 1e, 2b–g and 2i–q, 3b and 3d, 4a–e, 5b–f, 6b–f, 7d and 7f, 8b–f, 9g, and Supplementary Figs. 1, 4, and Supplementary Table 1 are provided as a source data file.

## Code availability

The MATLAB functions used for data analysis of the cellular electrophysiology experiments ($I_f$ and AP recordings) have been deposited and are freely available at https://github.com/pgeschwill/NCOMMS-Paper. The code in the repository is associated with a GNU General Public License (GPL 3.0).

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

## Acknowledgements

We gratefully acknowledge the excellent technical work of Simone Bauer and Ulrike Mersdorf. This work was supported in parts by grants from the Medical Faculty of the University of Heidelberg and the German Cardiac Society (postdoc fellowship and research scholarship to P.Y.), from the German Heart Foundation (Kaltenbach scholarship to T.F.), from the German Cardiac Society and the Hengstberger Foundation (Klaus-Georg and Sigrid Hengstberger Scholarship to D.T.), from the Joachim Siebeneicher Foundation (to D.T.), from the Deutsche Forschungsgemeinschaft (SCHW 1611/1-1 to P.A.S. and TH 1120/8-1 to D.T.), from the Max-Planck-Society (TANDEM project to P.A.S. and M.K.O.) and from the German Centre for Cardiovascular Research (DZHK) (to S.N., C.S., A.P.S., H.E., D.T., H.A.K., P.A.S.). S.N. is recipient of a DZHK doctoral student scholarship. P.A.S. is recipient of the Heidelberg Research Center for Molecular Medicine (HRCMM) Senior Career Fellowship.

## Author contributions

P.Y., M.K.O. and P.A.S. conceived and designed the study and the experiments. P.Y. and M.K.O. generated the transgenic *mouse* model. P.Y., S.N. and P.A.S. performed the molecular biology experiments. P.Y. and P.A.S. performed the histopathology and immunohistochemistry experiments of *mouse* hearts. P.G., C.B., C.S., A.P.S. and H.E. performed and analyzed the patch-clamp experiments. M.M., M.W. and R.F. carried out the calcium recordings of isolated cardiomyocytes. S.N., M.K., T.F. and P.A.S. performed the telemetric ECGs of *mice* and data analysis. S.N. and P.A.S. performed the *mouse* echocardiography and data analysis. H.A.K. provided administrative support and final approval of the paper. A.D. and D.T. supervised data analysis and provided expertise with paper writing. The paper was written principally by P.A.S.

## Additional information

**Competing interests:** The authors declare no competing interests.

