## [Peer Review File · Nature Communications]

Reviewers' comments:

Reviewer #1 (expert in cardiac HCN4)

Remarks to the Author:

This manuscript investigates the effects of HCN4 overexpression in working cardiomyocytes by generation of a transgenic mouse model where expression of hHCN4 channel subunits was controlled by the murine cardiac troponin I promoter.

The rationale of the study is well explained and the results are interesting and novel. Transgenic mice have dilated ventricular chambers and reduced wall thickness associated with higher basal Ca_i concentration and higher Ca_i transients, indicating a clear correlation with HCN4 overexpression.

While the main message of the work is outlined properly, the impact of the experimental findings is however limited by a flaw in the interpretation of data.

The authors interpret the increase in Ca_i resting levels and transients in cells isolated from transgenic mice as due to permeation of Ca^{++} ions through HCN4 channels, and in doing so refer to a 2004 PNAS paper (Yu et al) on the neuronal pacemaker current. However, Ca^{++} permeation is far from being an established property of f/HCN channels. Early work on Ca^{++} -dependence of the I_f current in cardiac cells by Hagiwara and Irisawa (J Physiol 1989, 409:121) showed that increasing intracellular Ca^{++} in whole-cell experiments increased I_f , the opposite of what expected from Ca^{++} permeation, and was interpreted to indicate Ca_i -dependent channel activation. This paper incidentally reported no change in the I_f reversal potential upon changes of Ca_i , indicating no permeation. Subsequent work in inside-out patches (Zaza et al Pflugers Arch 1991, 419:662) showed no direct Ca_i -dependent activation, suggesting involvement of Ca -activated kinases and cAMP, again with no evidence of permeation. The problem with Ca permeation is that existing evidence is only indirect. Data by Yu et al could be explained by Ca_i changes being secondary to local I_f -dependent Na_i changes and caused by variation in the Na-Ca exchange. The authors consider the NCX mechanism only in relation to the issue of exacerbation, but not to the issue of how the Ca_i increase is generated.

Finally, even admitting that the hypothesis of Ca permeation holds, it would not stand up to a simple calculation. I_f is a tiny current. With a diastolic depolarization <0.1 V/s and a cell capacity of say 24 pF (Mangoni & Nargeot 2001, Cardio Res. 52:51), the diastolic inward current of mouse pacemaker myocytes is about 2.4 pA. If according to Yu et al, only 0.6% of this current is carried by Ca^{++} , it would mean that the Ca component during diastole is of the order of 0.015 pA. This current would be grossly undersized to modify to any degree the cytosolic Ca concentration. Ca currents, more likely to be involved in changes of Ca_i transients, are in the range of 800 pA (Mangoni & Nargeot). The ratio between hypothetical Ca component of I_f and peak Ca current is therefore in the range of 20 millionths.

While the manuscript presents convincing evidence for the dependence between HCN4 expression and Ca homeostasis, it is highly unlikely that this is due to Ca permeation through HCN4 channels. The authors should investigate the nature of Ca_i changes and verify if these are for example associated with changes of Na_i gradient.

Special

- p. 3, l. 10: "pointing towards mechanisms of If blockade that are particularly beneficial in failing hearts". These data can also be interpreted as caused to high dosage administration (up to 10 mg bid) and contrasts with previous data from the Beautiful trial.

-figure 1f: the If current around -60 mV is negligible (point at -60 is even slightly positive). This evidence is hardly compatible with the idea that Ca_i changes are due to an inward Ca-component of If.

-figure 2e: "HCN4 overexpression did not significantly change the efficiency of Ca^{2+} release, reflected by the slope (e)" This does not explain why there is a significant decrease in the presence of ivabradine.

Reviewer #2 (expert in HCN4)

Remarks to the Author:

The manuscript by Yampolski et al investigates the consequences of overexpression of hHCN4 channels in the adult mouse heart. To this aim, the authors employed transgenic (TG) mice in which a hHCN4 construct is expressed under the control of the cTN1 promoter. Expression appears to be restricted to the working myocardium. The authors show that expression of hHCN4 channels dysregulates intracellular Ca^{2+} [Ca^{2+}]_i homeostasis and provokes cardiac hypertrophy and cellular apoptosis. The authors report that adverse effects of hHCN4 overexpression in TG mice are suppressed by ivabradine. The authors suggest that If block by ivabradine in failing hearts is a mechanism promoting favourable outcome of patients with heart failure treated with ivabradine. They also suggest that block of Ca^{2+} entry via ventricular If channels can explain heart-rate independent protection against ischemic injury by ivabradine. The approach is of interest but several

aspects of the study should be implemented with new data and/or clarified in relation to a complex pathology such as heart failure. My suggestions are as follows:

- The authors measured I_f in voltage-clamped TG ventricular myocytes and show robust current expression. Control data are not shown yet it would have been useful here, because it has been reported that I_f is expressed in adult ventricular myocytes, but its activation curve is shifted to very negative voltages. Expression of hHCN4 in TG hearts shows I_f at voltages close to that routinely recorded in the sino-atrial node, implying that overexpression of hHCN4 channels overcomes the well-known "context-dependence" of the I_f activation curve, which predicts that when heterologous HCN channels are expressed in a neonatal myocyte the activation curve will be shifted positive to the myocyte resting potential, while when expressed in an adult myocyte they will activate negative to the resting potential. The authors should accurately show control data and comment/discuss the literature on the context dependence of HCN expression and compare it to their own results.

- In Figure 1-2, I miss a detailed electrophysiological study of the ventricular action potential of control and TG myocytes. Indeed, Figure 1 shows robust I_f expression already at voltages positive to the normal myocyte resting potential. It can be expected that TG hHCN4 channels depolarize the normal resting potential, possibly leading to early diastolic depolarization of the membrane voltage in basal conditions or under beta-adrenergic activation. The authors state that the heart rate of TG mice does not differ from that of control mice however, this does not preclude the possibility of ventricular arrhythmia, which is an important associated risk factor in heart failure (and protection by ivabradine). In my view these data would add to the significance of the manuscript. Indeed, there exist reports of alteration in action potential of ventricular myocytes re-expressing HCN2. Protection from arrhythmia following ivabradine treatment can also explain positive outcome in HF patients.

- In Figure 2, the authors highlight increased basal $[Ca^{2+}]_i$ in TG myocytes and attribute this phenomenon to increased basal Ca^{2+} entry through TG hHCN4 channels. This points to my previous comment: if there exist Ca^{2+} entry through HCN4 channels it means that TG I_f contributes to the membrane voltage by being open at rest and very likely, during the repolarization phase of the action potential. In Figure 2A, I do not see increased basal $[Ca^{2+}]_i$. Are traces normalized? An increased $[Ca^{2+}]_i$ transient in TG myocytes suggests increased sarcoplasmic reticulum Ca^{2+} load. Is SR Ca^{2+} load increased? Is the cell size of TG myocytes normal?

- The authors first suggested that increased Ca^{2+} entry via HCN4 channels can alter Ca^{2+} homeostasis, but then they seem to favor increased Na^+ entry and reverse functioning of the NCX mode as the mechanism for $[Ca^{2+}]_i$ alteration. Both mechanisms imply that I_f contributes to the membrane voltage, which is not investigated. At least part of your effects on $[Ca^{2+}]_i$ homeostasis and cell death can be caused by dysregulated membrane excitability induced by hHCN4 conductance overexpression (so the protective effect of ivabradine). In conclusion: what happens to membrane voltage and Ca^{2+} permeability when you overexpress hHCN4?

-Two points merit further discussion. (1) You show that TG mice present hallmarks of hypertrophy and increased cell death via activation of caspase -dependent pathway. However, you did not find tissue fibrosis, which is reminiscent of $[Ca^{2+}]_i$ toxicity induced by hHCN4 overexpression. To which extent hHCN4 TG mice constitute a model of heart failure, a model of cell death accompanied with hypertrophy. (2) As you said, expression of I_f in the ventricle of failing heart has been reported. How

does the very strong expression of hHCN4 current in your TG mice compare to previous work on If in heart failure?

Reviewer #3 (expert in Ca²⁺ and ion channels in the heart/cardiac hypertrophy)

Remarks to the Author:

These authors engineered mice that express a transgene coding for HCN4, driven by the cardiac troponin I promoter. High level expression of transcript, protein, membrane-localized protein, and "funny current" (If) were observed. These animals developed cardiomyopathy, marked by ventricular dilatation and wall thinning, contractile dysfunction. Alterations in intracellular Ca handling were noted, as were increases in TUNEL-positive cells and caspase activation.

The stated rationale for this work derives from reports of up-regulated If in failing ventricular myocytes. Normally, the current is essentially undetectable, and it has been reported that If increases 2-3-fold in models of heart failure. Here, over-expression is much greater. The quantified data say 8-fold (Fig 1c), but it appears to me to be even higher than that (Fig 1c, 1e, 1f). It comes as little surprise to me that high level overexpression of a functional ion channel provokes a significant phenotype. Further, I do not believe this level of overexpression can be informative of the human disease context where increases are substantially less.

The technical aspects of the paper are strong, focusing mainly on electrophysiology and Ca handling. Strikingly missing from these data are echocardiographic measures of ventricular size and function.

Minor Comments:

"If current" is redundant.

To the Editorial Office

Submission number: NCOMMS-15-11664-T

“Responses to Reviewers”

Dear Sir or Madam,

Thank you for the review of our manuscript, which we have now carefully revised according to the remarks and suggestions of the reviewers.

We resubmit the revised manuscript to "**Nature Communications**", indicating our response to the comments made by the referees:

Reviewer #1:

“This manuscript investigates the effects of HCN4 overexpression in working cardiomyocytes by generation of a transgenic mouse model where expression of hHCN4 channel subunits was controlled by the murine cardiac troponin I promoter.

The rationale of the study is well explained and the results are interesting and novel. Transgenic mice have dilated ventricular chambers and reduced wall thickness associated with higher basal C_{ai} concentration and higher C_{ai} transients, indicating a clear correlation with HCN4 overexpression.

While the main message of the work is outlined properly, the impact of the experimental findings is however limited by a flaw in the interpretation of data.

The authors interpret the increase in C_{ai} resting levels and transients in cells isolated from transgenic mice as due to permeation of Ca^{++} ions through HCN4 channels, and in doing so refer to a 2004 PNAS paper (Yu et al) on the neuronal pacemaker current. However, Ca^{++} permeation is far from being an established property of f/HCN channels. Early work on Ca^{++} -dependence of the I_f current in cardiac cells by Hagiwara and Irisawa (J Physiol 1989, 409:121) showed that increasing intracellular Ca^{++} in whole-cell experiments increased I_f , the opposite of what expected from Ca^{++} permeation, and was interpreted to indicate C_{ai} -dependent channel activation. This paper incidentally reported no change in the I_f reversal potential upon changes of C_{ai} , indicating no permeation. Subsequent work in inside-out patches (Zaza et al Pflugers Arch 1991, 419:662) showed no direct C_{ai} -dependent activation, suggesting involvement of Ca -activated kinases and cAMP, again with no evidence of permeation.

The problem with Ca permeation is that existing evidence is only indirect. Data by Yu et al could be explained by C_{ai} changes being secondary to local I_f -dependent N_{ai} changes and caused by variation in the Na - Ca exchange. The authors consider the NCX mechanism only in relation to the issue of exacerbation, but not to the issue of how the C_{ai} increase is generated.

Finally, even admitting that the hypothesis of Ca permeation holds, it would not stand up to a simple calculation. I_f is a tiny current. With a diastolic depolarization <0.1 V/s and a cell capacity of say 24 pF (Mangoni & Nargeot 2001, Cardiovasc Res. 52:51), the diastolic inward current of

mouse pacemaker myocytes is about 2.4 pA. If according to Yu et al, only 0.6% of this current is carried by Ca^{++} , it would mean that the Ca component during diastole is of the order of 0.015 pA. This current would be grossly undersized to modify to any degree the cytosolic Ca concentration. Ca currents, more likely to be involved in changes of Ca_i transients, are in the range of 800 pA (Mangoni & Nargeot). The ratio between hypothetical Ca component of I_f and peak Ca current is therefore in the range of 20 millionths.

While the manuscript presents convincing evidence for the dependence between HCN4 expression and Ca homeostasis, it is highly unlikely that this is due to Ca permeation through HCN4 channels. The authors should investigate the nature of Ca_i changes and verify if these are for example associated with changes of Na_i gradient.”

We thank the reviewer for her/his detailed suggestions and comments with respect to the mechanisms underlying the dependence between HCN4 expression and Ca^{2+} homeostasis, which helped us to improve the revised manuscript dramatically.

We agree with the reviewer that Ca^{2+} permeation through HCN4 channels, if it occurs at all, may have only minor influence and may hardly contribute to the effects observed in our model.

In cardiomyocytes, the close interrelation of intracellular Na^+ and Ca^{2+} homeostasis is mediated mainly by the electrogenic NCX current, known to be highly sensitive to changes of $[Na^+]_i$ (Pieske, B. & Houser, S. R., Cardiovasc Res 2003). Thus, an increase of $[Na^+]_i$ will switch the NCX equilibrium to reverse mode, thereby increasing intercellular Ca^{2+} and effectively shortening AP by its electrogenic action (Armoundas, A. A. et al. Circ Res 2003).

Following the suggestions of the reviewer we examined the hypothesis that HCN4 overexpression-mediated Na^+ influx increases $[Ca^{2+}]_i$ via reverse mode NCX activity (illustrated in Figure 10). In agreement with this assumption, we observed that instant application of ORM-10103, an agent that effectively inhibits reverse mode NCX (Jost, N. et al. Br J Pharmacol. 2013; Nagy, N. et al. Br J Pharmacol. 2014) abolished Ca^{2+} overload in HCN4^{tg/wt} cardiomyocytes, demonstrating that NCX function is sensitive to increased HCN4 activity in our model. Likewise, inhibition of HCN4 channel activity by ivabradine reduced Na^+ influx to wildtype levels and adjusted NCX equilibrium towards the ‘forward mode’ leading to restored $[Ca^{2+}]_i$, similar to ORM-10103. Of note, depolarization of RMP by augmented I_f may lower the driving force of forward NCX in addition, thereby aggravating accumulation of $[Ca^{2+}]_i$ in HCN4^{tg/wt} hearts.

Furthermore we observed a shortening of AP duration in HCN4^{tg/wt} cardiomyocytes, mainly at APD20 and APD50, but this trend also remains at APD90. This finding strongly supports the notion that due to increased $[Na^+]_i$ ‘reverse mode’ NCX is activated in our model, eliciting a repolarizing driving force (according to Armoundas, A. A. et al. Circ Res 2003) that antagonizes the depolarizing effect of I_f at late repolarisation. This important and novel link between HCN4 expression and Ca^{2+} homeostasis is thoroughly discussed in the revised manuscript (Discussion page 11 line 17 to page 13 line 6).

- p. 3, l. 10: "pointing towards mechanisms of I_f blockade that are particularly beneficial in failing hearts". These data can also be interpreted as caused to high dosage administration (up to 10 mg bid) and contrasts with previous data from the Beautiful trial.

We agree with the reviewer that this is a speculation. As such, the sentence was modified in the introduction of the current manuscript (page 3 line 18) saying:

"... pointing towards mechanisms of I_f blockade that may be particularly beneficial in failing hearts."

-figure 1f: the I_f current around -60 mV is negligible (point at -60 is even slightly positive). This evidence is hardly compatible with the idea that Ca_i changes are due to an inward Ca-component of I_f .

The I_f current-voltage relationship has been re-evaluated by new experiments similarly recording ventricular cardiomyocytes isolated from HCN4^{tg/wt} and wild type mice. These new data are depicted in Figure 1e. With respect to the Ca^{2+} -permeation of HCN4 channels we agree with the Reviewer, as delineated above.

-figure 2e: "HCN4 overexpression did not significantly change the efficiency of Ca^{2+} release, reflected by the slope (e)" This does not explain why there is a significant decrease in the presence of ivabradine.

We thank the Reviewer for this relevant question. As we clarified above and in the manuscript, Ca^{2+} permeation through HCN4 channels, if it occurs at all, may have only minor influence on Ca^{2+} homeostasis; however, HCN4 overexpression-mediated Na^+ influx increased $[Ca^{2+}]_i$ via reverse mode NCX activity, as shown on Figure 6 and modeled on Figure 10. Therefore, we observed a similar effect incubating the HCN4^{tg/wt} cardiomyocytes with ivabradine or with ORM-10103 (pharmacological agent that inhibits reverse mode NCX) reducing large Ca^{2+} transients recorded in HCN4^{tg/wt} cardiomyocytes to WT levels. In line, the rise of $[Ca^{2+}]_i$ leads to an augmented Ca^{2+} induced Ca^{2+} release mechanism represented by an increased slope (non-significantly) resulting in a rise of the peak, given that the time to peak is similar (as it is).

Conversely, these data suggest that blocking HCN4 channel with ivabradine or the reverse mode of NCX with ORM-10103 reduce the excess of cytosolic Ca^{2+} (as observed on baseline of both figures 5 and 6) and thus reducing the Ca^{2+} induced Ca^{2+} release mechanism, thereby decreasing the slope in the presence of ivabradine or ORM-10103 (as it is).

We would like to clarify to the Reviewer that we have exchanged the slope parameter shown on Figure 2e from the first submission to the time to peak now shown on Figure 5e. Since there was only non-significant increase of the slope parameter in TG, going in parallel with a significant

increase of the peak, but a reverse change of both parameters in the presence of ivabradine or ORM-10103 we assume that the constancy of the time to peak parameter most precisely delineates the situation, given that higher peak is caused by increased slope and vice versa. Both slope and time to peak parameters describe the time relation of the Ca^{2+} release. In the interest of clarity and simplicity, we therefore decided to exchange to time to peak, because it is a more common and understandable parameter to the general audience than slope.

Reviewer #2:

The manuscript by Yampolski et al investigates the consequences of overexpression of hHCN4 channels in the adult mouse heart. To this aim, the authors employed transgenic (TG) mice in which a hHCN4 construct is expressed under the control of the cTN1 promoter. Expression appears to be restricted to the working myocardium. The authors show that expression of hHCN4 channels dysregulates intracellular Ca^{2+} $[\text{Ca}^{2+}]_i$ homeostasis and provokes cardiac hypertrophy and cellular apoptosis. The authors report that adverse effects of hHCN4 overexpression in TG mice are suppressed by ivabradine. The authors suggest that If block by ivabradine in failing hearts is a mechanism promoting favourable outcome of patients with heart failure treated with ivabradine. They also suggest that block of Ca^{2+} entry via ventricular If channels can explain heart-rate independent protection against ischemic injury by ivabradine. The approach is of interest but several aspects of the study should be implemented with new data and/or clarified in relation to a complex pathology such as heart failure. My suggestions are as follows:

- The authors measured If in voltage-clamped TG ventricular myocytes and show robust current expression. Control data are not shown yet it would have been useful here, because it has been reported that If is expressed in adult ventricular myocytes, but its activation curve is shifted to very negative voltages. Expression of hHCN4 in TG hearts shows If at voltages close to that routinely recorded in the sino-atrial node, implying that overexpression of hHCN4 channels overcomes the well-known "context-dependence" of the If activation curve, which predicts that when heterologous HCN channels are expressed in a neonatal myocyte the activation curve will be shifted positive to the myocyte resting potential, while when expressed in a adult myocyte they will activate negative to the resting potential. The authors should accurately show control data and comment/discuss the literature on the context dependence of HCN expression and compare it to their own results.

We thank the reviewer for her/his valuable comments regarding the electrophysiology of our model. According to the suggestions, we have now performed new experiments and have revised the manuscript to highlight the strengths of our study more precisely. Recording ventricular cardiomyocytes isolated from $\text{HCN4}^{\text{tg/wt}}$ and wild type (WT) mice in parallel confirmed the presence of I_f in WT ventricular myocytes and provided activation voltages in WT myocytes similar to values previously reported in mice (Graf, E. M. et al. Naunyn Schmiedebergs Arch Pharmacol. 2001), showing a shift to negative voltages compared to the activation in a sinoatrial nodal cell (Mesirca, P. et al. Nat. Com. 2014), as mentioned by the Reviewer.

In contrast, current expression in $\text{HCN4}^{\text{tg/wt}}$ cardiomyocytes is enlarged at membrane voltages $< -60\text{mV}$, with a moderate (2-3 fold) increase of I_f at physiological resting membrane potentials of

ventricular myocytes (~ -90mV) compared to WT cardiomyocytes. Interestingly, these changes reflect the magnitude of current increase recorded in cardiomyocytes of heart failure patients (Stillitano, F. et al. J Mol Cell Cardiol 2008). Thus, endogenous overexpression of HCN channels in mice elicit robust I_f that activates positive to the myocyte resting membrane potential. These findings were confirmed by HCN2 overexpressing transgenic mice showing an increased I_f in ventricular cardiomyocytes at voltages < -60mV (Kuwabara, Y. et al. J Am Heart Assoc. 2013). However, the activation properties of I_f in HCN4^{tg/wt} ventricular cardiomyocytes differs remarkably from that of sinoatrial nodal cells (Mesirca, P. et al. Nat. Com. 2014), as activation positive to -60mV is negligible in HCN4^{tg/wt} ventricular cardiomyocytes, which may indicate that the "context-dependence" of the I_f activation curve is in parts preserved.

- In Figure 1-2, I miss a detailed electrophysiological study of the ventricular action potential of control and TG myocytes.

Indeed, Figure 1 shows robust I_f expression already at voltages positive to the normal myocyte resting potential. It can be expected that TG hHCN4 channels depolarize the normal resting potential, possibly leading to early diastolic depolarization of the membrane voltage in basal conditions or under beta-adrenergic activation. The authors state that the heart rate of TG mice does not differ from that of control mice however, this does not preclude the possibility of ventricular arrhythmia, which is an important associated risk factor in heart failure (and protection by ivabradine). In my view these data would add to the significance of the manuscript. Indeed, there exist reports of alteration in action potential of ventricular myocytes re-expressing HCN2. Protection from arrhythmia following ivabradine treatment can also explain positive outcome in HF patients.

Thank you for this suggestion. We now have recorded action potential (AP) properties of ventricular cardiomyocytes isolated from HCN4^{tg/wt} and WT mice and have performed a detailed electrophysiological analysis, which is depicted in Figure 8 of the revised manuscript. We now show that transgenic hHCN4 channels change the morphology of the AP and, as expected by the Reviewer, depolarize the resting potential. In addition we have undertaken comprehensive experiments on cellular and *in vivo* arrhythmogenicity of HCN4^{tg/wt} cardiomyocytes/animals following the Reviewers advice.

Interestingly, we observed that myocytes, isolated from HCN4^{tg/wt} hearts, showed a higher incidence of spontaneous APs than WT cells, and that a subset of cells exerted virtual pacemaker function. Moreover, HCN4^{tg/wt} cardiomyocytes were prone to afterdepolarizations resulting in trains of premature APs. However, during telemetric recordings HCN4^{tg/wt} animals revealed no sustained arrhythmias, although rising numbers of premature ventricular captures and few nonsustained VTs indicated an increased arrhythmogenicity compared to WT animals. Based on these data we agree with the Reviewer that inhibition of pro-arrhythmic effects, associated with augmented I_f , may contribute to positive outcome in heart failure patients treated with ivabradine.

- In Figure 2, the authors highlight increased basal $[Ca^{2+}]_i$ in TG myocytes and attribute this phenomenon to increased basal Ca^{2+} entry through TG hHCN4 channels. This points to my previous comment: if there exist Ca^{2+} entry through HCN4 channels it means that TG I_f contributes to the membrane voltage by being open at rest and very likely, during the repolarization phase of the action potential. In Figure 2A, I do not see increased basal $[Ca^{2+}]_i$. Are traces normalized?

We observed significantly increased $[Ca^{2+}]_i$ in cardiomyocytes isolated from HCN4^{tg/wt} hearts compared to WT. Instead of normalized traces in the previous manuscript we now show original traces in Figures 5 and 6. As addressed in response to Reviewer 1 we agree that Ca^{2+} permeation through HCN4 channels, if it occurs at all, may have only minor influence and hardly contributes to changes in Ca^{2+} homeostasis observed in our model. Notably, in cardiomyocytes close interrelation of intracellular Na^+ and Ca^{2+} homeostasis were shown to be mediated mainly by the electrogenic NCX current, known to be highly sensitive to changes of $[Na^+]_i$ (Pieske, B. & Houser, S. R., *Cardiovasc Res* 2003). It is broadly accepted that rise of $[Na^+]_i$ switches the NCX equilibrium to reverse mode, thereby increasing $[Ca^{2+}]_i$ and effectively shortening AP by its electrogenic action (Armoundas, A. A. et al. *Circ Res* 2003).

Accordingly, instant application of ORM-10103, an agent that effectively inhibits reverse mode NCX (Jost, N. et al. *Br J Pharmacol.* 2013, Nagy, N. et al. *Br J Pharmacol.* 2014), abolished Ca^{2+} overload in HCN4^{tg/wt} cardiomyocytes, demonstrating the pathophysiological relevance of dysregulated NCX in our model and excluding Ca^{2+} permeation through HCN4 as the critical mechanism. Of note, depolarization of RMP by augmented I_f (Figure 8c) may lower the driving force of forward NCX in addition, thereby aggravating accumulation of $[Ca^{2+}]_i$ in HCN4^{tg/wt} hearts. Moreover, we observed a shortening of AP duration in HCN4^{tg/wt} cardiomyocytes, mainly at APD20 and APD50, but this trend remains at APD90, underlining that 'reverse mode' NCX is activated in our model due to increased $[Na^+]_i$ (Armoundas, A. A. et al. *Circ Res* 2003) and elicits a repolarizing driving force, thereby antagonizing the depolarizing effect of I_f at late repolarisation.

An increased $[Ca^{2+}]_i$ transient in TG myocytes suggests increased sarcoplasmic reticulum Ca^{2+} load. Is SR Ca^{2+} load increased? Is the cell size of TG myocytes normal?

We thank the Reviewer for the interesting question. We addressed this question in our isolated cardiomyocyte preparation loaded with 10 μ M Fluo4-AM and puffing them with caffeine 10 mM. Differently from the electrical stimulation, where the Ca^{2+} induced Ca^{2+} release process occurs only during the membrane depolarization, the caffeine protocol is currently used to calculate the Ca^{2+} content in the SR. As the Reviewer will observe in the figure attached for this rebuttal (please see below), a representative caffeine-induced Ca^{2+} release peak was significantly higher on HCN4^{tg/wt} cardiomyocytes. This result suggests that the SR Ca^{2+} is higher on HCN4^{tg/wt}

cardiomyocyte and it supports the experimental data of higher Ca^{2+} transients upon electrical stimulation (Figures 5a and 5c). Regarding the size of the cardiomyocytes, we would like to refer to figure 3b, where is shown that $\text{HCN4}^{\text{tg/wt}}$ cardiomyocytes have a significantly higher cross sectional area. This anatomical result supports the data of higher Ca^{2+} transients recorded from $\text{HCN4}^{\text{tg/wt}}$ cardiomyocytes.

Figure: Quantification of the amplitude [peak] of the caffeine-induced Ca^{2+} transients, which serves as a measure for sarcoplasmic reticular Ca^{2+} content. **a**, representative Ca^{2+} transients. **b**, results of the experiment (TG [n=20]; WT [n=12] ; mean \pm s.d.; ** $P < 0.01$, $\text{HCN4}^{\text{tg/wt}}$ vs. WT; two-tailed Mann-Whitney test).

- The authors first suggested that increased Ca^{2+} entry via HCN4 channels can alter Ca^{2+} homeostasis, but then they seem to favor increased Na^{+} entry and reverse functioning of the NCX mode as the mechanism for $[\text{Ca}^{2+}]_i$ alteration. Both mechanisms imply that Ca^{2+} entry contributes to the membrane voltage, which is not investigated. At least part of your effects on $[\text{Ca}^{2+}]_i$ homeostasis and cell death can be caused by dysregulated membrane excitability induced by hHCN4 conductance overexpression (so the protective effect of ivabradine). In conclusion: what happens to membrane voltage and Ca^{2+} permeability when you overexpress hHCN4?

Thank you very much for this comment. In the revised manuscript, we provide new data showing that augmented myocardial hHCN4, cause diastolic Na^{+} influx that depolarizes the normal resting potential, indicating that overexpressed I_f contributes to membrane voltage. We agree with the reviewer that dysregulated membrane excitability may contribute to cell death and changed Ca^{2+} homeostasis and our experimental findings support this notion. Notably, the cardiac NCX current is highly sensitive to changes of intracellular Na^{+} and Ca^{2+} homeostasis (Pieske, B. & Houser, S. R., *Cardiovasc Res* 2003), and is known to switch to reverse mode, in response to increased $[\text{Na}^{+}]_i$. To decipher a contribution of this mechanism to the altered Ca^{2+} homeostasis in $\text{HCN4}^{\text{tg/wt}}$ cardiomyocytes, we blocked NCX current using ORM-10103, resulting in normalization of intracellular Ca^{2+} , demonstrating the pathophysiological significance of dysregulated NCX.

However, this does not exclude minor Ca^{2+} permeation through HCN4 channels. We agree with the view of Reviewer 1 that there is little evidence for the latter scenario, and if it occurs, our data suggest only a minor influence of this effect with respect to the changes of Ca^{2+} homeostasis in our model. Accordingly, the predominant pathophysiological role of NCX is thoroughly discussed in the revised manuscript (Discussion page 11 line 26 to page 13 line 6).

-Two points merit further discussion. (1) You show that TG mice present hallmarks of hypertrophy and increased cell death via activation of caspase-dependent pathway. However, you did not find tissue fibrosis, which is reminiscent of $[\text{Ca}^{2+}]_i$ toxicity induced by hHCN4 overexpression. To which extent hHCN4 TG mice constitute a model of heart failure, a model of cell death accompanied with hypertrophy. (2) As you said, expression of I_f in the ventricle of failing heart has been reported. How does the very strong expression of hHCN4 current in your TG mice compare to previous work on I_f in heart failure?

(1) To address this question we have performed echocardiographic examination of WT and $\text{HCN4}^{\text{tg/wt}}$ animals, assessing the extent of structural and functional changes in our transgenic model. Our data clearly denote adverse cardiac remodeling in $\text{HCN4}^{\text{tg/wt}}$ mice of 3 and 6 months of age. While changes at 3 months were characterized mainly by reduced wall thickness, at 6 months of age, however, $\text{HCN4}^{\text{tg/wt}}$ mice displayed cardiac dilation (Fig. 2h, i) with lower left ventricular wall thickness (Fig. 2n-p) and systolic dysfunction with reduced fractional area shortening compared to WT animals. Thus, we present a novel mouse model with augmented expression of myocardial I_f leading to a cardiomyopathy phenotype. We interpret the structural changes as a consequence of chronic derangement of Ca^{2+} homeostasis causing increased myocardial apoptosis, given the molecular hallmarks of cardiomyopathy and increased caspase-dependent cell death in tissue of $\text{HCN4}^{\text{tg/wt}}$ hearts. However, as pathophysiological changes in our genetic model are present from birth on and persist chronically, the phenotype may differ from the histopathological changes caused through acute $[\text{Ca}^{2+}]_i$ toxicity known to cause patches of fibrotic tissue degeneration.

(2) At physiological RMP of ventricular cardiomyocytes (~ -90mV) patch clamp recordings revealed only a modest increase of I_f (2-3 fold higher) in $\text{HCN4}^{\text{tg/wt}}$ cardiomyocytes isolated from TG mice. This reflects the magnitude of current increase that was recorded in cardiomyocytes of heart failure patients showing significant upregulation of ventricular I_f . We therefore propose that our transgenic mice constitute a suitable approach to model pathophysiological changes of the ventricular myocardium with respect to augmented I_f , as reported in failing hearts. Of note, increased myocardial I_f was observed in various pathological cardiac states like hypertrophy, myocardial infarction and end stage heart failure. Thus, our approach, to the best of our knowledge, constitutes the first animal model that provides insight into the pathomechanisms, which are caused by an augmented myocardial I_f .

Reviewer #3:

These authors engineered mice that express a transgene coding for HCN4, driven by the cardiac troponin I promoter. High level expression of transcript, protein, membrane-localized protein, and "funny current" (I_f) were observed. These animals developed cardiomyopathy, marked by ventricular dilatation and wall thinning, contractile dysfunction. Alterations in intracellular Ca handling were noted, as were increases in TUNEL-positive cells and caspase activation.

The stated rationale for this work derives from reports of up-regulated I_f in failing ventricular myocytes. Normally, the current is essentially undetectable, and it has been reported that I_f increases 2-3-fold in models of heart failure. Here, over-expression is much greater. The quantified data say 8-fold (Fig 1c), but it appears to me to be even higher than that (Fig 1c, 1e, 1f). It comes as little surprise to me that high level overexpression of a functional ion channel provokes a significant phenotype. Further, I do not believe this level of overexpression can be informative of the human disease context where increases are substantially less.

We thank the reviewer for this comment. Various authors have reported the expression of I_f in cardiomyocytes derived from healthy ventricular myocardium in mice and humans. To address the referee's criticism we have performed new experiments and recorded I_f of ventricular cardiomyocytes isolated from HCN4^{tg/wt} and wild type (WT) mice in parallel to compare the magnitude of current.

First, we confirmed the presence of robust I_f in cardiomyocytes from healthy WT mice. In this context, our experiments showed activation voltages in WT myocytes similar to values reported previously (Graf, E. M. et al. Naunyn Schmiedebergs Arch Pharmacol. 2001).

Second, I_f expression in HCN4^{tg/wt} cardiomyocytes showed only a moderate (2-3 fold) increase at physiological resting membrane potentials of ventricular myocytes (~ -90mV) compared to WT cardiomyocytes. These changes reflect the magnitude of current increase that was detected in cardiomyocytes of heart failure patients (Stillitano, F. et al. J Mol Cell Cardiol 2008) compared to healthy controls.

Third, the reviewer assumption that HCN4 protein levels in HCN4^{tg/wt} cardiomyocytes may exceed the 8:1 ratio based on low-resolution immunoblots presented in our previous manuscript. We apologize for the poor illustrations and present now repeated immunoblottings for quantitative analysis of HCN4 protein in the ventricular myocardium of HCN4^{tg/wt} transgenic and wild type mice using different protein load (10, 20, 30 µg) for better comparison (please refer to Figure 1c). The improved new data confirmed the previously reported ~ 8-fold increase of HCN4 protein in HCN4^{tg/wt} versus WT ventricles. However, in this context it is important to note that the amount of HCN4 protein does not linearly conduct into I_f activity.

Based on these results we would like to emphasize, that in contrast to the reviewer's assumption, the transgenic model does not excessively overexpress myocardial I_f, but exerts I_f augmentation to a moderate extent, reaching levels similar to those observed in heart failure in the human disease context (Stillitano, F. et al. J Mol Cell Cardiol 2008).

The technical aspects of the paper are strong, focusing mainly on electrophysiology and Ca handling. Strikingly missing from these data are echocardiographic measures of ventricular size and function.

We thank the reviewer for this consideration and now have performed echocardiographic evaluation of wild type and HCN4^{tg/wt} animals, assessing the extent of morphological and functional changes in our transgenic model as shown in Figure 2.

Minor Comments:

"If current" is redundant.

Thank you. We have corrected this redundancy in the manuscript.

Based on the suggestions of the reviewers, we clarified several important issues and emended misleading interpretation of our data. We kindly ask you to consider our revised manuscript for publication, and we look forward to your assessment of our work.

Yours truly,

Patrick A. Schweizer

Reviewers' comments:

Reviewer #1 (Remarks to the Author):

General

With this re-submission the authors reconsider the interpretation of data presented in the first version of the manuscript.

The revision is substantial and involves in particular the interpretation of the main experimental finding of this work, i.e. the increase in intracellular Ca^{2+} resting levels and transients in cells isolated from transgenic HCN4tg/wt mice.

In fact I am personally quite satisfied with this outcome since it fits exactly the suggestion I had made in my first review, i.e. that rather than assuming an unwarranted Ca permeability of HCN4/funny channels, the authors "should investigate the nature of Ca_i changes and verify if these are for example associated with changes of Na_i gradient"

Now I find with satisfaction that this is exactly what has happened. The authors show in this revised version that the Ca^{2+} overload is due to reverse mode NCX activity associated with the large increase in Na^+ entry caused by overexpression of HCN4 channels, and analyze the cellular and organ effects associated with Ca^{2+} overload as well as their clinical consequences. They also show that pharmacological inhibition of I_f prevents Ca_i overload and protects from ventricular remodeling.

The novelty of the results remains interesting as it was in the first version, and the rationale of the study is properly outlined. The proposed interpretation of the mechanism underlying Ca_i transient increase in isolated myocytes, and correlated dilated ventricular chambers and reduced wall thickness of transgenic mice, is satisfactorily supported by experimental results.

One general comment I have concerns data from the old literature in Purkinje fibres, which the authors may have missed, and which already report a correlation between I_f and the intracellular Na^+ concentration (Glitsch et al Pflugers Arch. 1986 406(5):464-471; Boyett et al J. Physiol. 1987 384:405-429). These data were the first independent confirmation of the Na permeability of I_f as originally described (DiFrancesco J. Physiol. 1981b: 377-393), and importantly in relation to the present work, they were the first demonstration of a direct link between intracellular Na^+ concentration and the size of I_f flowing. This has been overlooked by the authors and I suggest that these considerations are included.

Minor

-In the list of references Stillitano et al (2008) has been listed twice (#4 and #11).

-l. 133: “rather unchanged” is not really an accurate statement, do you perhaps mean “not significantly changed”?

-l. 160 and further on: “slope of Ca²⁺ uptake”

-l. 216: “...sustained trains of automaticity, driven by spontaneous Ca²⁺ transients in HCN4tg/wt cardiomyocytes...”. I would not use the term “driven”, since there is no proof that automaticity is caused by Ca transients. Actually, it is unlikely that spontaneous Ca²⁺ transients in HCN4tg/wt cells are associated with membrane depolarization and pacing, since as the authors demonstrate, the NCX works in reverse mode and the NTX current is therefore outward, not inward. The increased automaticity could simply result from the increased pacemaker current. The authors may want to elaborate a bit on these considerations.

-l. 324-325: “reversal potential (-35 mV for HCN4 at physiological conditions) 35”. This is an incorrect citation, the paper by Fenske et al (2011) is on HCN3, not HCN4. Similar problem also later “...70% of HCN4-mediated current remains after 1 second 35”. The authors should obviously quote data on HCN4 (for example Ishii et al JBC 1999 274(18): 12835–39).

-l. 330 and further on: “...analysis of HCN4 channel deactivation during 9-second pulses demonstrated that more than 70% of HCN4-mediated current remains after 1 second”. I do not follow this reasoning. How long the cell remains at voltages more negative than -35 mV has little or no effect on current deactivation. Most of deactivation occurs at depolarized voltages during the action potential, since at positive voltages the deactivation time constant is much shorter than at negative voltages.

Dario DiFrancesco

Reviewer #2 (Remarks to the Author):

The authors have substantially and properly revised the manuscript. They performed the experiments I suggested and duly included new discussion points. This is appreciated and I have now only minor points, as follows:

-In Fig. 1, the authors show that ivabradine reduces heart hypertrophy, via inhibition of ventricular overexpressed HCN4 mediated f⁻ current. I would expect ivabradine to reduce also heart rate. To which extent could we affirm that, beside block of I_f in the working myocardium, heart rate reduction also contributes to hypertrophy reduction? At lower heart rates, I would expect cytosolic Ca²⁺ overload to be reduced independently from ventricular I_f blockade. On the other hand, you

show that the action potential amplitude of TG myocytes to be unchanged at different pacing rates. This point could be more directly discussed.

- I suggest the authors to include the SR Ca²⁺ load experiment presented in the rebuttal to my questions in the revised Supplementary section, for the potential reader.

Matteo E. Mangoni

Reviewer #3 (Remarks to the Author):

No new comments

To the Editorial Office

Submission number: NCOMMS-15-11664B

“Responses to Reviewers”

Dear Sir or Madame,

Thank you for the review of our manuscript, which we have now carefully revised according to the remarks and suggestions of the reviewers, and which we think has improved the manuscript significantly.

We resubmit the revised manuscript to "**Nature Communications**". Please find attached our point-by-point response to the referees' comments in blue, with the changes indicated in the manuscript in blue:

Reviewer #1:

“One general comment I have concerns data from the old literature in Purkinje fibres, which the authors may have missed, and which already report a correlation between I_f and the intracellular Na^+ concentration (Glitsch et al Pflugers Arch. 1986 406(5):464-471; Boyett et al J. Physiol. 1987 384:405-429). These data were the first independent confirmation of the Na permeability of I_f as originally described (DiFrancesco J. Physiol. 1981b: 377-393), and importantly in relation to the present work, they were the first demonstration of a direct link between intracellular Na^+ concentration and the size of I_f flowing. This has been overlooked by the authors and I suggest that these considerations are included.”

We thank the reviewer for this valuable comment with respect to the existing data providing evidence for the correlation between I_f and the intracellular Na^+ concentration. According to the reviewers suggestion we have included the following paragraph to the discussion:

“The impact of I_f on $[Na^+]_i$, in relation to membrane potential has been demonstrated in sheep Purkinje fibres by voltage clamp recordings in early works (Glitsch et al. 1986; Boyett et al. 1987) after I_f was originally described (DiFrancesco et al. 1981). These studies showed that membrane hyperpolarization to below -60 mV significantly increased intracellular Na^+ activity, and related this increase to Na^+ influx through I_f , thus providing a direct link between $[Na^+]_i$ and the size of I_f .” (Discussion, page 12, line 5-9).

Minor

-In the list of references Stillitano et al (2008) has been listed twice (#4 and #11).

Thank you very much. We have now deleted the duplication.

-I. 133: “rather unchanged” is not really an accurate statement, do you perhaps mean “not significantly changed”?

We agree with the reviewer and have changed the sentence into:

“Other genes regulating calcium handling and cellular electrophysiology, however, showed transcript levels that were not significantly changed.” (Results, page 6 line 1).

-I. 160 and further on: “slope of Ca²⁺ uptake”

Thank you. The sentence is corrected. (Results, page 6, lines 27-28).

-I. 216: “...sustained trains of automaticity, driven by spontaneous Ca²⁺ transients in HCN4tg/wt cardiomyocytes...”. I would not use the term “driven”, since there is no proof that automaticity is caused by Ca transients. Actually, it is unlikely that spontaneous Ca²⁺ transients in HCN4tg/wt cells are associated with membrane depolarization and pacing, since as the authors demonstrate, the NCX works in reverse mode and the NTX current is therefore outward, not inward. The increased automaticity could simply result from the increased pacemaker current. The authors may want to elaborate a bit on these considerations.

Thank you very much for this remark. We agree that the increased automaticity most likely is the result of increased I_f, and that spontaneous Ca²⁺ transients are not causative for this phenomenon. As suggested by the reviewer we have changed the respective sentence – since it is part of the Results of the manuscript we have added an additional sentence to the discussion to elaborate on this point, as follows:

“Recordings of [Ca²⁺]_i transients and APs (Fig. 9a-g) frequently revealed sustained trains of automaticity in HCN4^{tg/wt} cardiomyocytes (Fig 9c), while such changes were only sparsely observed in wild type cells.” (Results, page 8, line 28 to page 9, line 2).

“...a subset of cells exerted virtual pacemaker function. The increased automaticity may originate from I_f augmentation and not from spontaneous Ca²⁺ cycling-mediated membrane depolarisation,

as the NCX equilibrium is shifted towards 'reverse mode' leading to an outward, hyperpolarizing current." (Discussion, page 13, lines 10-13).

-I. 324-325: "reversal potential (-35 mV for HCN4 at physiological conditions) 35". This is an incorrect citation, the paper by Fenske et al (2011) is on HCN3, not HCN4. Similar problem also later "...70% of HCN4-mediated current remains after 1 second 35". The authors should obviously quote data on HCN4 (for example Ishii et al JBC 1999 274(18): 12835–39).

Thank you for this remark. We have now replaced the citation with respect to the reversal potential of HCN4 to Ishii et al. JBC 1999, as suggested by the Reviewer. (Discussion, page 13, lines 4-5).

-I. 330 and further on: "...analysis of HCN4 channel deactivation during 9-second pulses demonstrated that more than 70% of HCN4-mediated current remains after 1 second". I do not follow this reasoning. How long the cell remains at voltages more negative than -35 mV has little or no effect on current deactivation. Most of deactivation occurs at depolarized voltages during the action potential, since at positive voltages the deactivation time constant is much shorter than at negative voltages.

Thank you for pointing this out. We have deleted the misleading sentence from the manuscript.

Reviewer #2:

The authors have substantially and properly revised the manuscript. They performed the experiments I suggested and duly included new discussion points. This is appreciated and I have now only minor points, as follows:

-In Fig. 1, the authors show that ivabradine reduces heart hypertrophy, via inhibition of ventricular overexpressed HCN4 mediated f- current. I would expect ivabradine to reduce also heart rate. To which extent could we affirm that, beside block of If in the working myocardium, heart rate reduction also contributes to hypertrophy reduction? At lower heart rates, I would expect cytosolic Ca²⁺ overload to be reduced independently from ventricular If blockade. On the other hand, you show that the action potential amplitude of TG myocytes to be unchanged at different pacing rates. This point could be more directly discussed.

Thank you for this valuable comment. We agree with the consideration of the Reviewer that the application of ivabradine did both: Antagonizing the effect of overexpressed ventricular I_f and reduction of sinoatrial nodal I_f, the latter causing a reduction of heart rate, as well.

We have now statistically evaluated the action potential amplitudes of HCN4^{tg/wt} myocytes at different pacing rates and found that the amplitudes significantly declined with higher pacing rates (Figure 8b). This is consistent with increased [Na⁺]_i at higher rates, in agreement with the consideration of the reviewer and previous literature (Pieske, B. & Houser, S.R., Cardiovasc Res

57, 874-886, 2003). Thus, increased heart rates may aggravate the maladaptive changes caused by the altered $\text{Na}^+/\text{Ca}^{2+}$ homeostasis in $\text{HCN4}^{\text{tg/wt}}$.

However, since cardiac remodelling/dilation in $\text{HCN4}^{\text{tg/wt}}$ mice developed at unchanged heart rates (Supplementary Figure 4, Supplementary Table 1), we conclude that rise of ventricular I_f is the predominant mechanism underlying the observed structural changes. Nonetheless, we agree with the Reviewer that the relevance of heart rate reduction for the normalisation of myocardial $\text{Na}^+/\text{Ca}^{2+}$ homeostasis and protection from remodelling needs to be addressed more explicitly in the discussion. This issue is now presented in the following sections of the revised manuscript:

“Noteworthy, the heart rate lowering effect of ivabradine may contribute to protection from adverse remodelling as well.” (Discussion, page 11, lines 16-17).

“In addition, $[\text{Na}^+]_i$ is known to rise in a rate dependent fashion³², which exacerbates $[\text{Ca}^{2+}]_i$ overload at high heart rates.” (Discussion page 12, lines 3-5).

[with respect to ivabradine]...”Synergistically, heart rate lowering decreases $[\text{Na}^+]_i$ ³² and therefore shifts the $\text{Na}^+/\text{Ca}^{2+}$ exchanger equilibrium towards the forward mode, which will alleviate intracellular $[\text{Ca}^{2+}]_i$ overload independently from ventricular I_f blockade.” (Discussion page 12, lines 20-22).

- I suggest the authors to include the SR Ca^{2+} load experiment presented in the rebuttal to my questions in the revised Supplementary section, for the potential reader.

Thank you very much for this remark. We have now included the SR Ca^{2+} load experiment as the new Supplemental Figure 1. The content is described under Results, as follows:

“In addition, Ca^{2+} load experiments using caffeine 10 mM (Supplementary Fig. 1) showed that the caffeine-induced Ca^{2+} release peak was significantly higher in $\text{HCN4}^{\text{tg/wt}}$ cardiomyocytes, indicating that the SR Ca^{2+} content is increased, which is consistent with the experimental data of higher Ca^{2+} transients upon electrical stimulation (Figures 5a and 5c).” (Results, page 7, lines 3-7).

Thank you very much. We look forward to your assessment of our revised manuscript.

Yours truly,

Patrick A. Schweizer

REVIEWERS' COMMENTS:

Reviewer #1 (Remarks to the Author):

“The impact of I_f on $[Na^+]_i$, in relation to membrane potential has been demonstrated in sheep Purkinje fibres by voltage clamp recordings in early works (Glitsch et al. 1986; Boyett et al. 1987) after I_f was originally described (DiFrancesco et al. 1981).”

This should be reworded.

The DiFrancesco (1981) paper (a one-author-only paper) is not the one where I_f was originally described, that was a 1979 paper in Nature (Brown DiFrancesco & Noble (1979) Nature 280, 235-236). The 1981 paper was the first to describe the mixed Na and K ionic nature of I_f , so the first to demonstrate a link between I_f and Na entry. I would suggest:

“...after the I_f mixed Na^+ and K^+ ionic nature was originally described (DiFrancesco, 1981).

Reviewer #2 (Remarks to the Author):

I have no further comments

To the Editorial Office

Submission number: NCOMMS-15-11664C

“Responses to Reviewers”

Dear Sir or Madame,

Thank you for the review of our manuscript, which we have now carefully revised according to the remark of the reviewer and the editorial requests.

We resubmit the revised manuscript to "**Nature Communications**". Please find attached our point-by-point response to the referees' comment in blue, with the changes indicated in the manuscript using the 'track changes' feature in Word:

Reviewer #1:

“The impact of I_f on $[Na^+]_i$, in relation to membrane potential has been demonstrated in sheep Purkinje fibres by voltage clamp recordings in early works (Glitsch et al. 1986; Boyett et al. 1987) after I_f was originally described (DiFrancesco et al. 1981).” This should be reworded. The DiFrancesco (1981) paper (a one-author-only paper) is not the one where I_f was originally described, that was a 1979 paper in Nature (Brown DiFrancesco & Noble (1979) Nature 280, 235-236). The 1981 paper was the first to describe the mixed Na and K ionic nature of I_f , so the first to demonstrate a link between I_f and Na entry. I would suggest:

“...after the I_f mixed Na^+ and K^+ ionic nature was originally described (DiFrancesco, 1981)”.

We thank the reviewer for this correction and have changed the sentence according to his suggestion.

(Discussion, page 13, lines 6-7)

Yours truly,

Patrick A. Schweizer